# Contractile injection systems facilitate sporogenic differentiation of *Streptomyces davawensis* through the action of a phage tapemeasure protein-related effector

Toshiki Nagakubo [1,2] ✉, Tatsuya Nishiyama[3], Tatsuya Yamamoto[1], Nobuhiko Nomura[1,2,4] & Masanori Toyofuku [1,2] ✉

Contractile injection systems (CISs) are prokaryotic phage tail-like nanostructures loading effector proteins that mediate various biological processes. Although CIS functions have been diversified through evolution and hold the great potential as protein delivery systems, the functional characterisation of CISs and their effectors is currently limited to a few CIS lineages. Here, we show that the CISs of *Streptomyces davawensis* belong to a unique group of bacterial CISs distributed across distant phyla and facilitate sporogenic differentiation of this bacterium. CIS loss results in decreases in extracellular DNA release, biomass accumulation, and spore formation in *S. davawensis*. CISs load an effector, which is a remote homolog of phage tapemeasure proteins, and its C-terminal domain has endonuclease activity responsible for the CIS-associated phenotypes. Our findings illustrate that CISs can contribute to the reproduction of bacteria through the action of the effector and suggest an evolutionary link between CIS effectors and viral cargos.

Contractile injection systems (CISs) are phage tail-like nanostructures with diverse functionalities. While contractile tails of phages play a crucial role in the translocation of viral genetic materials, known CISs act as nanomachines injecting CIS-associated effector proteins into target cells, resulting in various biological consequences.[1,2] Tailed phages, CISs, and type VI secretion systems, another group of phage tail-like contractile injection systems, share key structural features and action mechanisms,[3–5] suggesting that the tail structure has conferred a significant advantage and has been conserved during the divergence of these nanostructures from a common ancestor. Recently, *Photorhabdus* virulence cassette (PVC), a well-studied Gram-negative bacterial CISs, was demonstrated to serve as a programmable protein delivery system for animals.[6] While this study highlights the immense potential of CISs in various bioscience and biotechnology applications, our current understanding of their biological functions remains limited to only a few bacterial species. Therefore, expanding the repertoire of functionally characterised CISs is critical for the further exploration of the CIS potential. To date, several effector proteins associated with proteobacterial CISs have been identified as possessing enzymatic functions. Pnf was found as a PVC needle-associated effector protein with Rho-GTPase activity disrupting the cytoskeleton of target eukaryotic cells.[2,7] Pne1 was shown to be a DNA/RNA non-specific nuclease associated with metamorphosis-associated contractile structure (MAC), a CIS-related nanoparticle that stimulates the metamorphosis of a marine tubeworm.[8] Despite these insights, previous research on CIS functions has mainly focused on proteobacterial CISs, leaving vast room for further discovery of unique CISs and their cognate effector proteins.

Recent genomic studies have revealed a wide distribution of CIS-related gene clusters in prokaryotes.[4,5] In addition, several

[1]Faculty of Life and Environmental Sciences, University of Tsukuba, Tsukuba, Japan. [2]Microbiology Research Center for Sustainability (MiCS), University of Tsukuba, Tsukuba, Japan. [3]Life Science Research Center, College of Bioresource Sciences, Nihon University, Chiyoda, Japan. [4]Life Science Center for Survival Dynamics, University of Tsukuba, Tsukuba, Japan. ✉e-mail: nagakubo.toshiki.gp@u.tsukuba.ac.jp; toyofuku.masanori.gf@u.tsukuba.ac.jp

evolutionary lineages of CIS gene clusters have been proposed.[5] Members of each lineage share synteny and are generally conserved among phylogenetically closely related bacterial groups.[5] While most of the CIS gene clusters are thought to be acquired from ancestors by the vertical transfer, in some cases, structurally similar CIS gene clusters are conserved among phylogenetically distant species. Although these unique and widely shared CIS gene clusters are intriguing sources of CISs with novel functionalities, they have not yet been investigated.

*Streptomyces* species are members of gram-positive actinobacteria and are known for their remarkable abilities to produce various secondary metabolites including therapeutic antibiotics. The typical *Streptomyces* life cycle encompasses the following morphological differentiations: spore germination, vegetative growth of substrate mycelia by apical tip extension, and, in response to environmental conditions, the erection of aerial mycelia which finally turn into spores.[9] During this life cycle, *Streptomyces* species establish multicellular networks of mycelia and then form dispersal unicellular spores, thereby maintaining the colony-wide fitness.[10] In addition to these unique characteristics of reproduction, *Streptomyces* species have attracted considerable interests because of the high conservation of CIS-related gene clusters within the genus. Very recently, several studies have shown that intracellularly localised CISs produced by model *Streptomyces* species can affect the morphological differentiation, presumably through different mechanisms depending on the species and/or growth conditions.[11–14] While these observations suggest that *Streptomyces* CISs have been maintained in the genomes by directly or indirectly affecting the fitness of the producer bacteria, in our preliminary bioinformatics analysis, we noticed that CIS-related gene clusters with low similarities to the typical *Streptomyces* CISs can be found in the genomes of several non-model *Streptomyces* species. One of these *Streptomyces* species is *Streptomyces davawensis* JCM 4913 that was originally isolated from a Philippine soil sample and is known to produce the antibiotic roseoflavin.[15]

In this study, we show that the *S. davawensis* CISs belong to a unique group of CISs distributed across distant bacterial phyla and facilitate aerial mycelia erection and spore formation of the producer bacterium in dense culture. This phenotype involves increases in extracellular DNA (eDNA) release and biomass accumulation. An endonuclease effector interacting with CIS baseplate proteins was identified as being responsible for these CIS-associated phenotypes. The CIS effector shares homology with phage tapemeasure proteins which the importance in tail length determination and translocation of genetic materials has been suggested in tailed phages. Our findings highlight the unprecedented role of CISs in the reproductive life cycle of the producer bacterium and shed light on the viral origin of the effector-dependent CIS functionalities.

## Results

### A putative CIS is encoded within the Streptomyces davawensis genome

Phylogenetic analysis revealed that the amino acid sequence of a putative CIS sheath protein of *S. davawensis* JCM 4913 belongs to a clade which includes CIS proteins from unusually diverse bacterial phyla including Proteobacteria and Chloroflexota, besides Actinobacteria (Fig. 1a, b; Supplementary Figs. 1 and 2). Notably, this clade of CIS gene products is distant from the clades of structurally and/or functionally investigated CISs, especially CIS$^{Sc}$ and SLP (*Streptomyces lividans* phage tail-like nanostructure), which are members of the largest class of actinobacterial CISs (Fig. 1b).[11,14] Synteny analysis of these CIS gene clusters further showed the high diversity of the host bacteria in terms of their taxonomy and cell structures (diderm or monoderm) despite the structural similarities of the gene clusters (Supplementary Fig. 2). One of these gene clusters was located in the plasmid (Supplementary Fig. 2). Overall, these bioinformatic analyses suggest that

the CIS-related gene cluster of *S. davawensis* represents a unique group of uninvestigated CISs that may have been widely transmitted among phylogenetically distant bacterial species.

### CISs increase spore formation of Streptomyces davawensis in dense culture

To investigate the CISs of *S. davawensis*, we first attempted to delete the CIS-related genes of this bacterium via homologous recombination. Unfortunately, this attempt was unsuccessful possibly due to low efficiency of the homologous recombination at the genomic locus. We thus introduced a CRISPR-Cas9n-based genome editing system expressing the fusion enzyme APOBEC-Cas9n-UGI that enables conversion of specific cytosine upstream of the protospacer adjacent motif (PAM) sequence into thymine, thereby creating a stop codon within the gene of interest.[16] Although the initial version of the system (pCRISPR-cBEST)[16] was not stably maintained in *S. davawensis* probably due to the high basal expression of APOBEC-Cas9n-UGI, we constructed a modified, fine-tuneable editing system and finally succeeded in knocking out of the CIS-related genes. We replaced the ribosome-binding site downstream of the leaky thiostrepton-inducible promoter ($P_{tipA}$) of pCRISPR-cBEST with a theophylline-inducible riboswitch which strictly regulates the translation of the downstream open reading frame in a theophylline concentration-dependent manner (Fig. 1c; Supplementary Fig. 3).[17] Using this modified system, we optimised the editing conditions and converted Gln78 (CAG) of a gene encoding a putative sheath protein (CisS) of CISs into a stop codon (TAG) terminating its translation (Fig. 1c). We then prepared cell lysates of this knockout mutant (ΔcisS) and the wildtype strain and observed the ultracentrifugation pellets of the lysates by transmission electron microscopy. Rod-like structures were observed only in the wildtype lysate and their sizes are comparable with those of typical phage tails (Fig. 1d).[18] On the other hand, we could observe granule-like structures in the lysates of ΔcisS mutant, which might be unmatured CIS particles (Fig. 1d). We thus assumed that the rod-like structures observed in the wildtype lysate would be the CIS particles encoded by the CIS-related gene cluster.

Next, we first cultivated the *S. davawensis* strains on mannitol-soya flour (MS) medium normally used for *Streptomyces* cultures. While there were no apparent differences in spore formation between the strains in low density cultures (approximately $10^2$ viable spores per plate), we found that aerial mycelia erection and spore formation were severely impaired in the ΔcisS mutant under high density culture condition (approximately $10^5$ viable spores per plate) (Fig. 1e, f; Supplementary Fig. 4). This phenotypic defect in the ΔcisS mutant was restored by in trans complementation with *cisS* (Fig. 1e, f). Therefore, these results suggest that the loss of CISs decreases spore formation of *S. davawensis* in highly dense cultures whereas they are not essential for the cellular processes of morphological differentiation.

### CIS loss alters multicellular structures and extracellular matrix compositions in an early S. davawensis colony

To investigate how CISs affect the phenotype of *S. davawensis* in the dense cultures, we measured the growth of the wildtype strain and the ΔcisS mutant. Since growth of these strains in typical liquid media was very slow and highly variable, we inoculated spores on a cellophane placed onto a solid MS medium and measured biomass after cultivation. Under this modified growth condition, the dense cultures of the ΔcisS mutant showed impaired aerial mycelia erection, which is consistent with the phenotype of this strain grown without the cellophane (Figs. 1f and 2a–c). Overall, the growth of *S. davawensis* on the cellophane membrane could be divided into two phases: phase 1 (day 0–4) and phase 2 (day 5–8) (Fig. 2a). Notably, during the phase 1, a statistically significant decrease in growth was detected in the ΔcisS mutant, suggesting that CISs impacted the *S. davawensis* growth at an early growth stage (Fig. 2a; Supplementary Fig. 5). Bright-field and confocal

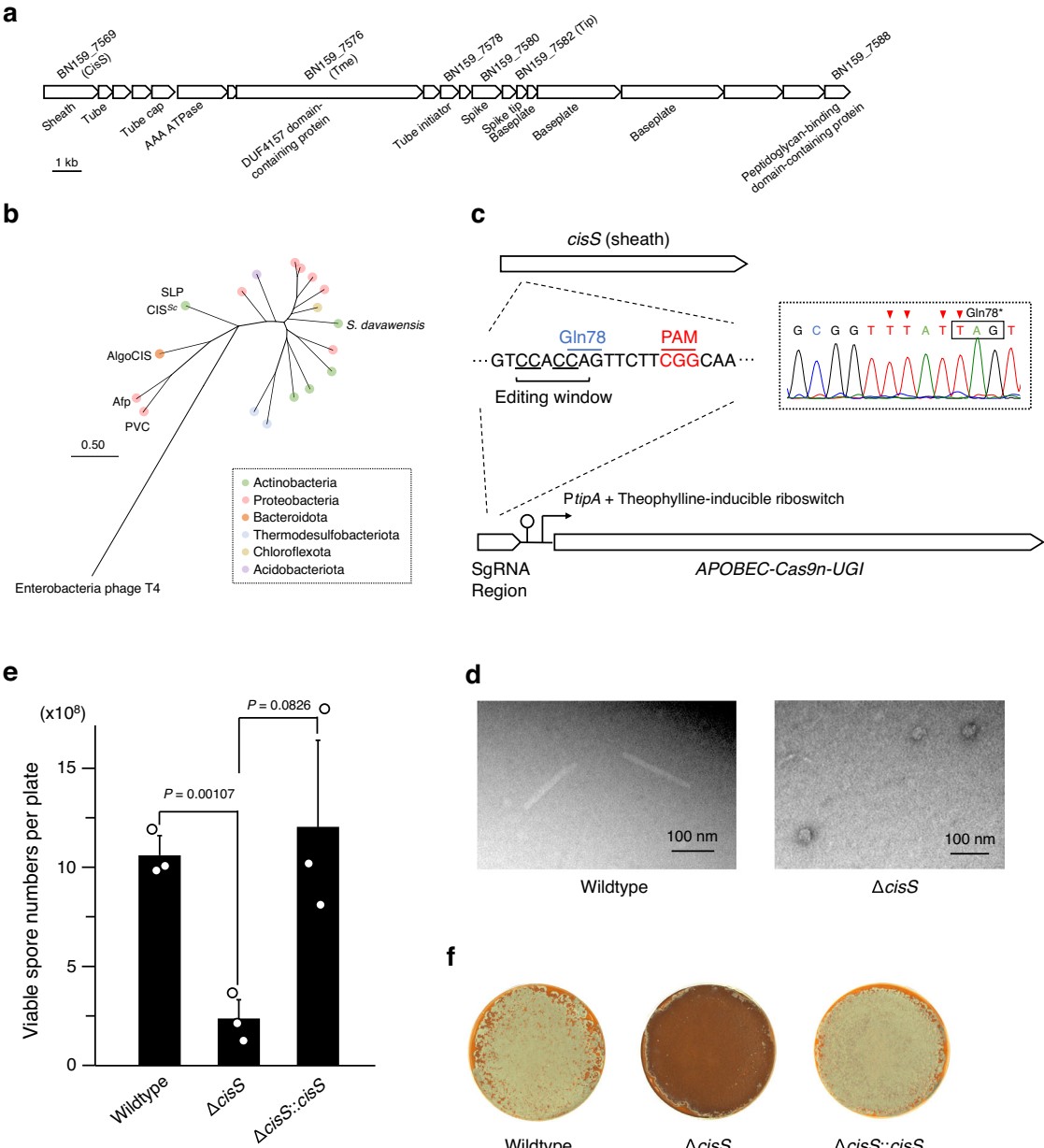

**Fig. 1 | Impact of CISs on spore formation of *S. davawensis*. a** Gene cluster encoding CIS structural proteins in *S. davawensis* JCM4913 with gene locus tags (BN159_XXXX) from the complete genome sequence (accession: HE971709.1) and gene annotations. **b** Phylogenetic tree of CIS sheath proteins constructed using the Maximum Likelihood method and JTT matrix-based model. The tree was drawn to scale, with branch lengths in the same units of the evolutionary distances used to infer the phylogenetic tree. CIS gene cluster organisations closely related to *S. davawensis* CISs are shown in Supplementary Fig. 2. Proteins used for this analysis are listed in Supplementary Table 4. **c** CRISPR-Cas9n-based gene knockout scheme for *S. davawensis*, targeting *cisS*. The target sequence within *cisS* was synthesised and introduced into the single guide RNA (sgRNA) region of the pCRISPR-cBEST-RS plasmid. PAM, protospacer adjacent motif. Knockout of *cisS* was confirmed by sequencing (represented by a square with dotted lines), with edited nucleotides indicated by red arrowheads. **d** Representative transmission electron microscopy images of the CIS extracts are shown. These images show the extracts from the wildtype strain and the Δ*cisS* mutant. **e** Comparison of spore formation rates under high density culture condition among *S. davawensis* strains. Approximately $10^5$ viable spores were spread onto MS medium and incubated at 30 °C for 2 weeks. *cisS* was integrated at the *attC* site. Bars represent mean ± S.D. for three independent cultures. *P* values were calculated using two-sided *t*-test with Welch's correction. **f** Morphologies of colonies used in panel (**e**). Representative images for three independent cultures of each strain are shown. The white regions of colony surfaces indicate aerial mycelia erection. Source data are provided as a Source Data file.

microscopy analyses revealed that the surface of the Δ*cisS* mutant colonies in phase 1 was rather flat compared to that of the wildtype colonies and apparently lacked mycelia erecting into the air (Fig. 2d, e; Supplementary Fig. 6). In addition, the substrate mycelia of the Δ*cisS* mutant inside the colony were densely packed within biofilm-like multicellular structures, whereas the wildtype formed mesh-like multicellular networks with wider intercellular spaces (Fig. 2f; Supplementary Fig. 6). In many cases, such multicellular architectures of

bacterial colonies are associated with the compositions of the extracellular matrix (ECM) that cement cells together and critically affect the organisation of densely populated microbial communities including *Streptomyces*.[19–22] We thus assumed that the distinct multicellular structures of *S. davawensis* strains may reflect differences in ECM composition. Therefore, we quantified extracellular DNA (eDNA), saccharides, and proteins, which are the major constituents of typical bacterial ECM, in colonies of *S. davawensis* strains. Notably, the

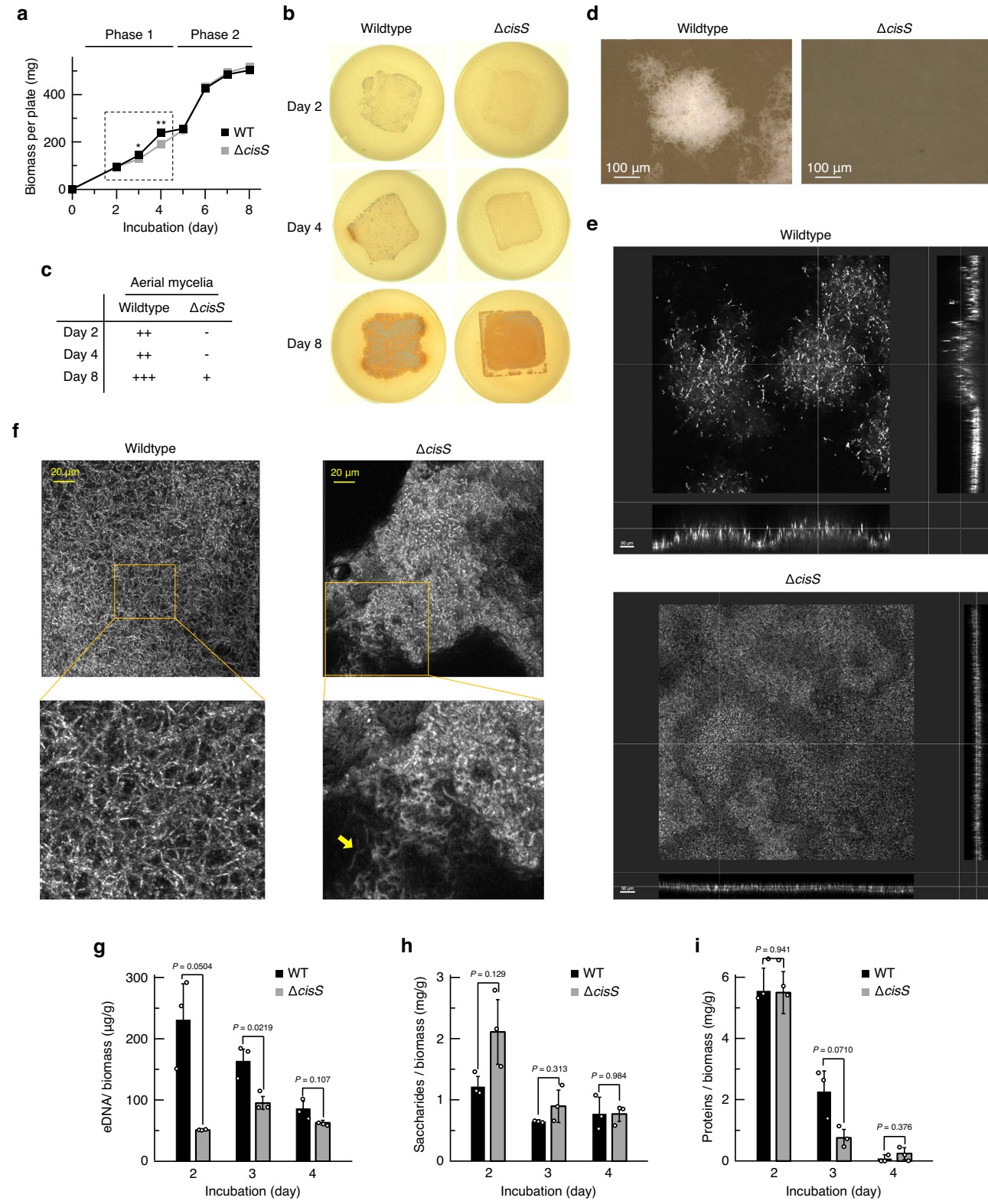

**Fig. 2 | CISs affect multicellular structures and extracellular matrix compositions of *S. davawensis* colony.** The colony phenotypes of the *S. davawensis* strains were analysed. **a** Growth curves of the *S. davawensis* strains are shown. The square with broken lines indicates colony growth during phase 1. Values represent mean values ± S.D. for three independent cultures. *P* values were calculated using two-sided *t*-test with Welch's correction. *, *P* = 0.0733. **, *P* = 0.0134. **b** Colony morphologies at the designated time points are shown. Representative images for three independent cultures of each strain are shown. **c** Aerial mycelia formation at the designated time points was determined from the appearance of white regions in the colonies. **d** Colony surfaces of the *S. davawensis* strains were observed by bright-field microscopy. Colonies were grown for 2 days. Aerial mycelia (white) were erected from wildtype colonies. Scale bars, 100 μm. **e, f** Representative reflection confocal images of the *S. davawensis* colonies on day 2 are shown. The microscopy settings are described under the Methods. The yellow arrow in panel f indicates substrate mycelia entangled in the biofilm-like structure. Scale bars in panel e, 50 μm. **g–i** Extracellular matrix constituents were quantified. The detailed procedures are described under the Methods. Bars indicate mean ± S.D. for three independent cultures. *P* values were calculated using two-sided *t*-test with Welch's correction. Source data are provided as a Source Data file.

**a**

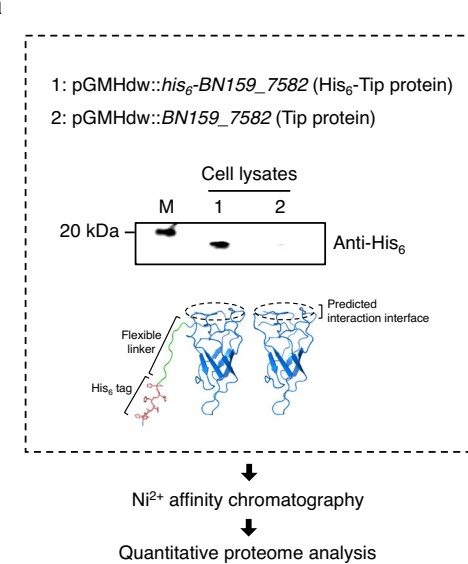

1: pGMHdw::*his₆-BN159_7582* (His₆-Tip protein)
2: pGMHdw::*BN159_7582* (Tip protein)

**b**

| Detected protein | Abundance Ratio | Gene ID |
|---|---|---|
| Predicted tube initiator protein | 5.38 | BN159_7578 |
| Predicted spike protein | 2.63 | BN159_7580 |
| ATP synthase subunit beta | 1.65 | BN159_3038 |
| Metal-binding protein | 1.49 | BN159_6661 |
| Elongation factor Tu | 1.45 | BN159_3731 |
| DUF4157 domain-containing protein | 1.34 | BN159_7576 |
| Chaperone protein DnaK | 1.24 | BN159_4524 |
| NAD-dependent glyceraldehyde-3-phosphate dehydrogenase | 1.17 | BN159_6535 |
| 30S ribosomal protein S2 | 1.15 | BN159_2785 |
| Chaperonin GroEL | 1.08 | BN159_3657 |

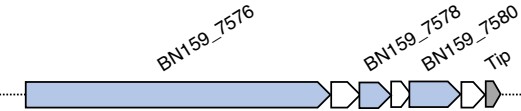

**Fig. 3 | Putative DUF4157 domain-containing effector is associated with *S. davawensis* CISs.** Identification of a putative effector of *S. davawensis* CISs. **a** *S. davawensis* strains expressing Tip or His₆-Tip under control of the constitutive *hrdB* promoter were constructed. The tagged-Tip was detected by western blot using an anti-His₆ antibody. Proteins extracted from 200 µg (wet weight) of the colonies were loaded into each lane. The AlphaFold2-predicted structures of Tip and His₆-Tip are also shown. Circles with broken lines indicate regions predicted to interact with a VgrG-like spike protein. The lysates of the *S. davawensis* strains were subjected to Ni²⁺ affinity chromatography and subsequent proteomic analysis. **b** Proteins significantly enriched in the His₆-Tip eluates (q-value < 0.05) are listed. Three biologically independent samples were analysed for each strain. The abundance ratio indicates the abundance of the detected proteins in the His₆-Tip eluates relative to that in the Tip eluates. The CIS-related proteins are highlighted in blue. To note, Tip was not included in this list because of the absence of peptides detectable by the mass spectrometer used in this study. Source data are provided as a Source Data file.

extractable ECM materials of the wildtype colonies were considerably enriched in eDNA, especially on day 2 (Fig. 2g). In contrast, the extractable extracellular saccharides were more abundant in the ΔcisS mutant colonies (Fig. 2h). Protein concentrations in the ECM extracts were more variable between the time points than the other constituents and, unlike the other ECM materials, did not show significant differences on day 2 (Fig. 2i).

It has been reported that a substantial amount of genomic DNA is released as eDNA within bacterial colonies.[23] In *S. davawensis*, a band of high molecular weight DNA (>10 kbp), which would be derived from genomic DNA, was detected only in the wildtype ECM on day 2, accounting for the higher eDNA concentration in this sample (Fig. 2g; Supplementary Fig. 7). This suggests the involvement of CISs in the release of genomic DNA-derived eDNA from *S. davawensis* mycelia.

### BN159_7576, a putative DUF4157 domain-containing effector, is associated with CISs

As CISs typically exhibit their biological functions through ejecting effector proteins, we hypothesised that *S. davawensis* CISs load and eject specific effector protein(s) that mediate the CIS-associated phenotypes. To identify the putative CIS-associated effector protein(s), we constructed a strain expressing CISs tagged with hexahistidine at the PAAR-repeat protein-like BN159_7582 (Tip) which is predicted to be a needle tip-like protein that forms a complex with a VgrG-like spike protein (Figs. 1a and 3a).[24,25] A strain expressing a non-tagged Tip was also constructed as a control. Subsequent quantitative proteomic analysis of Ni²⁺ affinity chromatography eluates of mycelial lysates in the early growth phase revealed that CIS baseplate-related proteins (a VgrG-like spike protein BN159_7580 and a probable tube initiator protein BN159_7578) and a DUF4157 domain-containing protein (BN159_7576) were significantly enriched in the His₆-Tip eluates, indicating that these proteins were coeluted with the His₆-Tip protein during the affinity chromatography (Fig. 3b). In previous works,

DUF4157 domain has been identified as a core domain of various candidate CIS effectors.[26] Therefore, our proteomic analysis suggests that during the early growth phase, the putative DUF4157 domain-containing effector physically interacting with the CIS baseplate proteins was ejected from the CIS particles.

### Characterisation of BN159_7576 as a tapemeasure protein-related endonuclease

Putative effector BN159_7576 is a large protein with unique structural characteristics. Its secondary structure includes hydrophobic regions, which are predicted to form multiple transmembrane helices, and probable disordered regions accounting for approximately 70% of the polypeptide (Fig. 4a). Probable coiled-coil segments were also predicted (Fig. 4a; Supplementary Fig. 8). Furthermore, profile hidden-Markov model-based sequence analysis indicated that BN159_7576 has, in addition to the above-mentioned DUF4157 domain, two distinct regions showing remote homology with either phage tapemeasure proteins or DNA entry endonucleases including EndA of *Streptococcus pneumoniae* (Fig. 4a). DNA entry endonucleases are associated with the cell envelope and have been proposed to promote the transfer of DNA polymers across the cellular membrane via partial hydrolysis of the DNA substrates.[27] Tapemeasure proteins have been found in tailed phages, and their polypeptide lengths have been observed to correlate with the lengths of the phage tails.[28] Based on the homology between BN159_7576 and tapemeasure proteins indicating a potential evolutionary relationship (Supplementary Fig. 9), we have assigned the name "tapemeasure-related effector" (Tme) to BN159_7576. The C-terminal domain of Tme and EndA from *S. pneumoniae* share several key structural characteristics essential for the catalytic function. Specifically, both the DRGH and H-N-N motifs, which include a proposed base catalyst histidine and Mg²⁺-coordinating asparagine, respectively, of EndA-related nucleases are conserved in Tme (Fig. 4b).[29] In addition, the predicted structure of the Tme C-terminal domain and the crystal

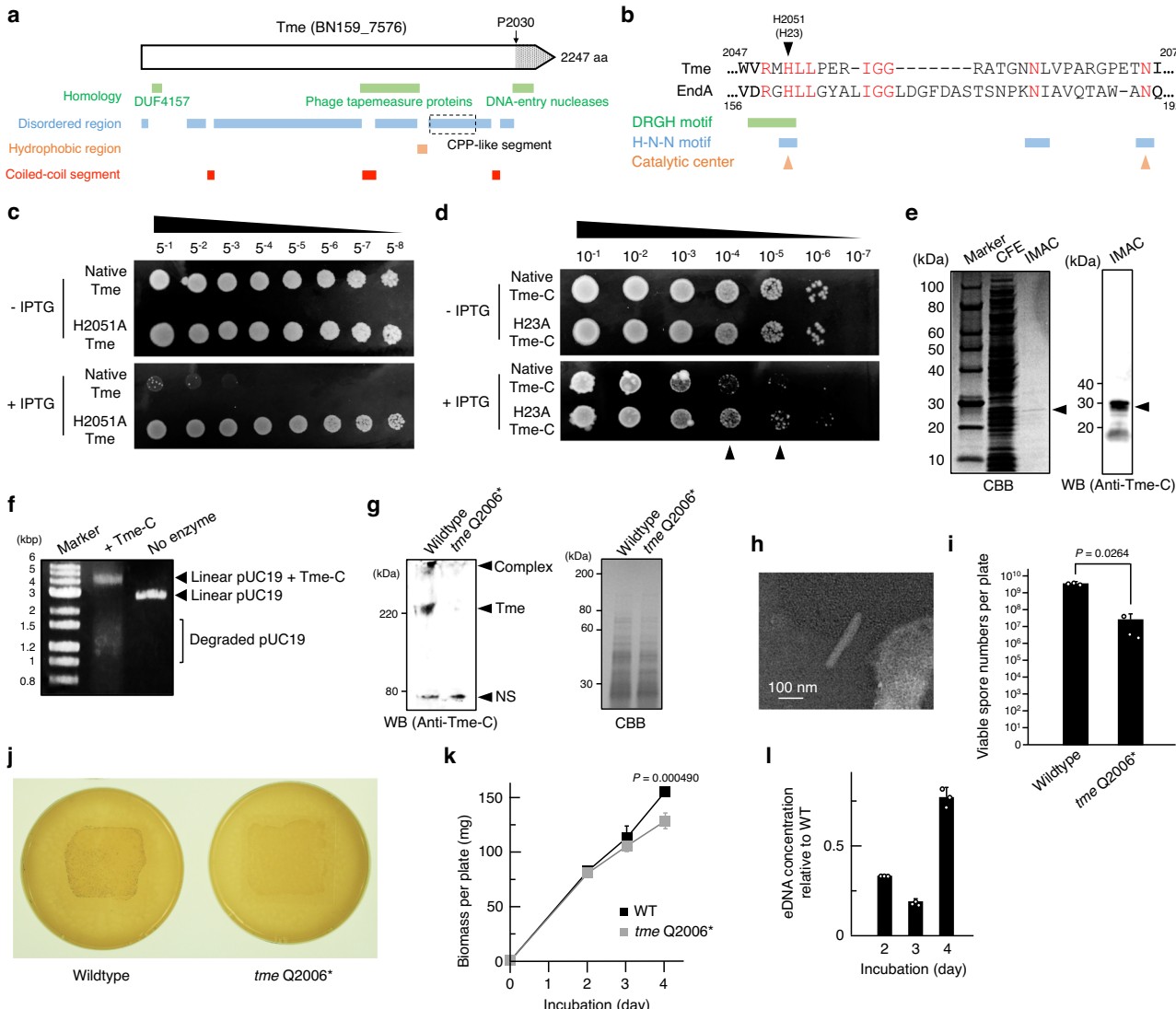

**Fig. 4 | Endonuclease activity of a tapemeasure protein-like effector is responsible for the CIS-associated phenotypes of *S. davawensis*.** The C-terminal domain of the putative CIS effector Tme was characterised as the endonuclease domain responsible for the CIS-associated phenotypes of *S. davawensis*.

**a** Secondary structure characteristics of Tme are shown. P2030 corresponds to the amino acid from which the sequence of the truncated form of Tme (Tme-C) begins after an additional sequence at the N-terminus. A cell-penetrating peptide (CPP)-like segment is indicated by a square with dashed lines. **b** Partial sequence of the C-terminal domain of Tme was aligned with *Streptococcus pneumoniae* EndA. Conserved amino acids are shown in red. H2051 and H23 correspond to the predicted catalytic histidine of the full length Tme and its truncated form Tme-C, respectively. **c**, **d** Cytotoxic effects of Tme and Tme-C overexpression on *E. coli* BL21 was evaluated. Protein expression from pET26b::*tme*, pET26b::*tme* (H2051A) pET26b::*tme-C-his₆*, and pET26b::*tme-C* (H23A)-*his₆* was induced by adding 1 mM isopropyl-β-D-thiogalactopyranoside (IPTG) to LB-Lennox medium. Serially diluted cultures of *E. coli* BL21 harbouring each of the pET26b derivatives were incubated at 30 °C for 2 (Tme-C) or 3 (Tme) days. Black arrowheads in panel **d** indicate the dilution rates at which His23-dependent cytotoxicity was detected. **e** His₆-tagged Tme-C (Tme-C-His₆) was isolated from the cell lysate of *E. coli* BL21 harbouring pET26b::*tme-C-his₆*. CFE, cell-free extract. IMAC, eluates of immobilised-metal affinity chromatography on a Ni²⁺-Sepharose column. CBB, Coomassie Brilliant Blue staining. WB, western blotting using anti-Tme-C serum. Black arrowheads indicate the bands of Tme-C-His₆. **f** Nuclease activity of the isolated Tme-C was assayed. pUC19 was linearised by SmaI-XbaI digestion. The reaction conditions are described under the Methods. All reaction mixtures were mixed with 0.05% (w/v) SDS and

then subjected to agarose gel electrophoresis. To note, DNA concentration in the reaction mixtures were increased to 5 ng to detect degradation products. The reaction mixtures were incubated for 1 h. **g** Lysates of the *S. davawensis* strains were analysed by SDS–PAGE and western blotting. Mycelia were grown for 3 days. Proteins extracted from 200 μg (wet weight) of the colonies were loaded into each lane. CBB, Coomassie Brilliant Blue staining. WB, western blotting using anti-Tme-C serum. NS, non-specific band derived from an endogenous protein of *S. davawensis*. Complex, an SDS-resistant complex containing Tme. The band above 220 kDa corresponds to the full length Tme. **h** Representative transmission electron microscopy image of the extracted *tme* Q2006* CIS particles is shown. **i** Spore formation rates of the *S. davawensis* strains were compared. Culture conditions were the same as those described in Fig. 1d. Bars represent mean values ± S.D. for three independent cultures. *P* values were calculated using two-sided *t*-test with Welch's correction. **j** Colony morphologies of the *S. davawensis* strains are shown. Colonies were grown on cellophane membranes placed onto MS medium for 2 days. Representative images for three independent cultures of each strain are shown. **k** Growth curves of the *S. davawensis* strains are shown. The measurement conditions were the same as those in Fig. 2a. Values represent mean values ± S.D. for five independent cultures. *P* values were calculated using two-sided *t*-test with Welch's correction. **l** eDNA concentrations in the extracellular matrix (ECM) extracts were compared between the *S. davawensis* strains. eDNA concentrations in the ECM extracts were normalised to biomass and the normalised values are shown relative to those of the wildtype culture at the designated time point. Bars represent mean values ± S.D. for three independent cultures. Source data are provided as a Source Data file.

structure of *S. pneumoniae* EndA (PDB entry: 3OWV) share a similar active site conformation created by two crossing strands and one helix (Supplementary Fig. 10).[29] Considering these structural features, we hypothesised that the C-terminal domain of Tme might have nucleolytic activity. To test this hypothesis, we first introduced either the pET26b::*tme* plasmid or its derivative with a His2051Ala mutation into *E. coli*. While the overproduction of native *tme* caused cytotoxicity, the mutated *tme* had no detectable effect on *E. coli* growth (Fig. 4c). Furthermore, the circumvention of cytotoxicity by alanine substitution at the corresponding histidine (His23) was also observed for the truncated C-terminal domain of Tme (Tme-C; 2030-2247 of Tme) (Fig. 4d). Relatively lower toxicity of Tme-C to *E. coli* would suggest that the N-terminal DUF4157 domain and/or the predicted transmembrane helices enhanced the toxicity.[30] For a more detailed characterisation of the Tme-C activity, we isolated His$_6$-tagged Tme-C from *E. coli* lysates (Fig. 4e). Tme-C formed a probable covalent complex with a linear double-stranded DNA substrate, leading to a significant upward shift of the band in agarose gel, and degraded the substrate DNA (Fig. 4f). Tme-C with a His23Ala mutation did not exhibit nucleolytic activity, indicating that the toxic effect of overexpressed Tme and Tme-C detected in the in vivo assay could be attributed to their nucleolytic activity (Supplementary Fig. 11). Furthermore, this result also suggests the involvement of His23 in the complex formation and rules out the possibility that contaminating *E. coli* enzyme(s) degraded the substrate.

*S. pneumoniae* EndA is a Mg$^{2+}$-dependent endonuclease that degrades both double- and single-stranded DNA and belongs to the DNA/RNA non-specific endonuclease family.[27] As Tme-C can also degrade supercoiled pUC19 as well as its linearised form, it is unlikely to have exo-specific nucleolytic activity (Supplementary Fig. 12a, b). In addition, the nucleolytic activity was inhibited by the addition of chelators for divalent cations and then restored by further addition of excess of Mg$^{2+}$ (Supplementary Fig. 12c). Furthermore, Tme-C degraded single-stranded DNA and RNA (Supplementary Fig. 12d, e). These results collectively suggest that Tme-C shares, at least in part, a common catalytic mechanism with EndA. However, the formation of a probable covalent complex has not been reported for EndA that the native enzyme could not be isolated from *E. coli* cells due to high toxicity of its nucleolytic activity.[27] Alanine mutation of Glu55 of Tme-C, which corresponds to Glu205 crucial for the EndA-catalysed hydrolytic reaction, seemed to inhibit the formation of a probable covalent complex with 20 min of incubation (Supplementary Notes; Supplementary Figs. 10b and 13).[29] These observations suggest that the active site residues of Tme-C are involved in not only hydrolytic reaction but also the formation of a probable covalent complex, potentially limiting hydrolytic activity of Tme-C to a significantly lower level compared with EndA-related endonucleases in which the active site residues would be dedicated to hydrolysing the DNA phosphodiester (Supplementary Notes; Supplementary Fig. 13). Moreover, several amino acids facilitating hydrolysis of the DNA phosphodiester by EndA are not conserved in Tme-C (Supplementary Notes; Supplementary Fig. 14). These data may imply a mechanism limiting the nucleolytic activity of Tme-C and provide a possible explanation why Tme-C could be isolated from the *E. coli* cells. Currently, we are investigating the catalytic mechanism of Tme-C in detail.

### Endonuclease activity of the cell envelope-localised endonuclease Tme would be responsible for the CIS-associated phenotypes of S. davawensis

As described above, the C-terminal domain of the candidate effector Tme was characterised as a CIS-associated endonuclease. As genomic DNA would be the natural substrate for this enzyme, we reacted the extracted genomic DNA with the isolated Tme-C. After incubation for 3 h, the size of the genomic DNA apparently decreased, indicating partial hydrolysis of genomic DNA by endonuclease activity

(Supplementary Fig. 7). The size of the degraded genomic DNA was comparable to that of the high molecular weight band observed only in the wildtype ECM (Supplementary Fig. 7), raising the possibility that the band appearing in the ECM sample could be Tme-degraded genomic DNA. We also analysed Tme localisation in *S. davawensis* colony by western blot analysis using anti-Tme-C antibody and found the lipid membrane-associated localisation of Tme, which is consistent with the presence of the hydrophobic regions within Tme (Fig. 4a; Supplementary Fig. 15). In addition, we noted that a segment located downstream of the hydrophobic regions within Tme is particularly rich in amino acids with basic side chains (arginine, 32%; lysine, 8.8%) (Fig. 4a). This segment spans 250 amino acids and exhibits a high theoretical isoelectric point of 11.57. Bioinformatic predictions indicated that the cationic segment possesses cell-penetrating peptide (CPP)-like properties (CPP probability 0.972540 on C2Pred).[31] Furthermore, a peptidoglycan-binding protein BN159_7588 is encoded at the end of the CIS gene cluster of *S. davawensis* (Fig. 1a). The genomic context of *BN159_7588*, the heterocomplex formation between BN159_7588 and a putative receptor-binding protein-related protein BN159_7587, and the binding of BN159_7588 to the *S. davawensis* mycelia suggested a possible role of BN159_7588 in CIS attachment to the *S. davawensis* cell envelope (Supplementary Fig. 16). These results collectively indicate that Tme is targeted to the cell envelope, potentially degrading genomic DNA and bridging intracellular and extracellular environments at the cellular membrane.

To gain insight into the biological significance of the enzymatic activity of Tme in *S. davawensis*, we constructed a mutant strain expressing endonuclease activity-less CISs using the riboswitch-dependent genome editing system targeting *tme* (Fig. 1c). Among the several editing points tested, only the Gln2006* stop mutation immediately upstream of the Tme-C domain was successful (Fig. 4a and g; Supplementary Fig. 17). CIS-like particles with ~200 nm length was observed in the lysate of the Gln2006* mutant (Fig. 4h). We also analysed the phenotypes of the Gln2006* mutant and the wildtype strain of *S. davawensis*. The spore formation rate of the mutant in dense culture was significantly lower than that of the wildtype strain (Fig. 4i). In addition, biomass, aerial mycelia formation, and eDNA concentration during phase 1 were significantly decreased by the Gln2006* mutation (Fig. 4j–l). Furthermore, a visible band of high molecular weight eDNA (>10 kbp) was not detected by gel electrophoresis of the mutant ECM (Supplementary Fig. 18). As all these characteristics of the Gln2006* mutant were similar to those of the Δ*cisS* mutant lacking mature CIS particles, the endonuclease activity of Tme would be responsible for the CIS-associated phenotypes of *S. davawensis*.

## Discussion

The inferred relationship between Tme and tapemeasure proteins of tailed phages provides evolutionary and functional insights into Tme as a new type of CIS effector. Tapemeasure proteins are typically the largest open reading frames in phage genomes, and in many cases, contain coiled-coil segment(s).[28] Besides their roles in determining tail lengths, they have also been implicated in the translocation of genetic materials across cellular membranes. It has been shown that tapemeasure protein of siphophages forms an initiator complex with the following proteins encoded by downstream genes: distal tail protein (Dit) and tail-associated lysin (Tal).[28,32] The initiator complex (tapemeasure protein-Dit-Tal) forms a hub around which the baseplate is constructed. The similar tapemeasure protein-baseplate complex is also formed in contractile myophages,[33] suggesting the generality of the assembly pattern. In the proposed cryo-EM structure-based model of genome injection by the *Staphylococcus aureus* phage 80α, initial binding of the phage onto cell surface subsequently allows for penetration of the plasma membrane by the Tal rod helices, which triggers release of the tapemeasure protein homologous to Tme

(Supplementary Fig. 9).[32] A series of in vitro and in vivo data have suggested that a pore-like structure that can be used for phage genome entry is formed within the membrane, probably by the extruded tapemeasure protein containing transmembrane helices.[34–37] Tme of *S. davawensis* CIS is highly likely to be evolutionarily related to tapemeasure protein of tailed phages for the following reasons: (*i*) *tme* is located immediately upstream of the baseplate-related genes and is the largest open reading frame in the CIS gene cluster (Fig. 1a), (*ii*) the amino acid sequence of Tme has remote homology with phage tapemeasure proteins (Fig. 4a; Supplementary Fig. 9), (*iii*) Tme contains the predicted multiple transmembrane helices and coiled-coil segments (Fig. 4a), and (*iv*) Tme would physically interact with the downstream spike protein and tube initiator protein encoded in the CIS gene cluster (Figs. 1a and 3b), forming a multiprotein complex analogous to the tapemeasure protein-baseplate complex of tailed phages.[25,32,33] Given these similarities of tapemeasure proteins and Tme, it can be speculated that Tme is also ejected from the cell envelope-attached CIS particles and is then inserted into the cellular membrane (Supplementary Figs. 15 and 16), possibly forming a pore-like structure. The endonuclease activity of Tme may be required to split genomic DNA into less bulky fragments, thereby allowing the fragments to pass through the putative pore, a receptor protein interacting with Tme, or other secretion system(s) into extracellular milieu. Additionally, as proposed for the related cell envelope-associated DNA entry endonucleases,[38] Tme may provide DNA termini through its nucleolytic action for efficient transfer of DNA polymers across the membrane. Moreover, the presence of the CPP-like segment within Tme supports the proposed model of Tme function in the translocation of molecules across lipid membranes, as the CPP-like segment could be involved in membrane destabilisation and the induction of pores across the phospholipid bilayers (Fig. 4a).[39,40] Since we could not detect a positive correlation between the concentrations of eDNA and proteins in the ECM extracts of the ΔcisS mutant (Fig. 2i), it is unlikely that the sheath-dependent action of the mature CIS particles directly induced cell death which ultimately results in non-specific leakage of the cellular materials via cell lysis. Low nucleolytic activity and spatial sequestration of Tme, and multinuclearity of *Streptomyces* mycelia would allow the producer bacterium to circumvent a physiologically detrimental consequence of the Tme action (Supplementary Notes).

*S. davawensis* CISs would affect the physical and chemical properties of the *S. davawensis* ECM and indirectly regulate the multicellular development of the bacterium (Fig. 2). Previous studies have suggested that eDNA is a major determinant of the viscoelasticity of the bacterial ECM,[41] and it is possible that the decrease in eDNA upon CIS loss alters the mechanical properties of the early *S. davawensis* colony (Fig. 2g). In addition, the relative increase in extracellular saccharide concentration may enhance ECM stability, potentially obstructing the escape of mycelia from multicellular aggregates (Fig. 2h).[19,20] CIS-associated changes in these ECM factors could result in the encasement of mycelia within abnormal multicellular structures that limit nutrient access and prevent mycelia from erecting into air (Fig. 2f). Additionally, as proposed for several *Streptomyces* species, eDNA may also be utilised as a redistributed nutrient and/or recyclable building blocks of DNA polymers for the reproductive development of *S. davawensis*.[42] The eDNA utilisation by bacteria often involves the degradation of high molecular weight eDNA into smaller fragments or monomers[43] and the presence of a similar eDNA digestion route in *S. davawensis* is implied by the extinction of high molecular weight eDNA during the growth (Supplementary Fig. 7). *S. davawensis* has a predicted extracellular DNase (*BN159_5097*), a close homolog of a functionally characterised extracellular DNase ExoSc of *Streptomyces coelicolor* A3(2) expressed during aerial mycelia formation.[44] Under high density culture conditions which limit nutrient availability and the physical space surrounding each colony, the physiological consequences of CIS-associated eDNA release would have a more

profound impact on the reproductive *S. davawensis* life cycle (Fig. 1f; Supplementary Fig. 4). Given the distinct compositions of the gene clusters, the *S. davawensis* CIS would be unique in its function among the investigated *Streptomyces* CISs, implying that there has been a niche-associated selective pressure driving the selection of CISs in *Streptomyces* (Fig. 1b; Supplementary Fig. 19; Supplementary Notes).

Our findings on the CIS-associated increase in eDNA release may also be relevant to the widespread distribution of certain CIS gene clusters. The CIS gene cluster of *S. davawensis* is a member of a unique group of CIS gene clusters containing *tme*-like genes, which are defined by the relative length of the open reading frame, the presence of the DUF4157 domain at the N-terminal region, and the conserved synteny within the gene clusters (Supplementary Fig. 2). This group of CIS gene clusters is conserved among highly diverse bacterial phyla (Fig. 1b; Supplementary Fig. 2), implying horizontal gene transfer events of the CIS gene clusters across bacterial ancestors. As reported in previous studies, once eDNA is released from cells, it can be internalised by other cells through various routes, and in some cases, environmental genetic information can be acquired when it provides benefits to the recipient bacteria.[45] Higher eDNA concentrations in microbial communities may potentially increase the probability of such evolutionary events, ultimately driving the acquisition and widespread distribution of genetic materials. Therefore, the CIS-associated increase of eDNA release observed in the present study may have contributed, at least in part, to the cross-species transmission of CIS information.

In summary, we provide novel insights into the biological function of *S. davawensis* CISs, a member of a widely distributed group of bacterial CISs. Our findings illustrate the unprecedented role of CISs in facilitating sporogenic differentiation of the producer bacterium through the enzymatic action of a phage-related CIS effector Tme. Tme-like CIS effectors may have diverged from phage tapemeasure proteins during the CIS evolution from a viral ancestor, and they now form a family of CIS effectors distributed across various bacterial phyla. In addition, the *S. davawensis* CISs and their relatives can potentially be applied as an alternative platform for programmable protein delivery systems through simple domain exchange at the C-terminus of Tme. Further investigation of the Tme-harbouring CISs will lead to the discovery of novel functionalities and promote their applications as versatile nanomachines.

## Methods

### Culture conditions

Strains used in this study are listed in Supplementary Table 1. *S. davawensis* was obtained from RIKEN BRC (Ibaraki, Japan). *S. davawensis* strains and *Streptomyces lividans* TK23 were routinely grown on mannitol-soya flour (MS) medium comprising 2 g mannitol, 2 g defatted soya flour, and 2 g agar per 1 L. For growth measurement, microscopic observation, and analysis of extracellular matrix, a sterilised cellophane membrane was placed onto MS medium and then spore solution (approximately $10^5$ viable spores) was spread onto the membrane. Biomass was measured by wet or dry weight of the colonies grown on the cellophane membrane at the designated time point. Soluble proteins were extracted from the above colonies by sonication in 10 mM HEPES-NaOH buffer (pH 7.4) and subsequent centrifugation at $7000\,g \times 5\,min$. *E. coli* strains were grown in LB-Lennox medium comprising 5 g yeast extract, 10 g tryptone, and 5 g NaCl per 1 L.

### Genetic manipulations

Primers and plasmids used in this study are listed in Supplementary Table 2. Ligation High Ver.2 (TOYOBO Co., Ltd., Osaka, Japan) and T4 polynucleotide kinase (Takara Bio Inc., Shiga, Japan) were used for DNA ligation. NEBuilder (New England Biolabs, MA, USA) and In-Fusion (Takara Bio Inc.) was used for sequence-independent DNA cloning. For gene knockout in *S. davawensis*, pCRISPR-cBEST plasmid[16] was

modified as follows. The flanking region of ribosome binding site-*APOBEC-Cas9n-UGI* was amplified by PCR (primer set cBEST_Inverse_Fw and Rv), and then fused with a synthetic theophylline-inducible riboswitch sequence by overlap extension PCR (primer set cBEST_Inverse_Fw, Thy_Riboswitch, and cBEST_Overlap_Rv). The resultant fragment was digested with NdeI and BamHI, and then ligated with *APOBEC-Cas9n-UGI* fragment which was cut from the original pCRISPR-cBEST plasmid using the same restriction enzymes. The resultant pCRISPR-cBEST-RS was digested with NcoI and then fused with DNA oligomers for single guide RNA (CisS_Q78_1 and 2; Tme_Q2006_1 and 2). Each of the plasmids was introduced into *E. coli* ET12567/pUZ8002. The transformed *E. coli* was grown in liquid LB-Lennox medium containing 20 μg/mL apramycin, 20 μg/mL kanamycin, and 5 μg/mL chloramphenicol, and the cells at the early exponential growth phase were collected from 1 mL of the culture. The collected *E. coli* cells were resuspended with *S. davawensis* spore solution (approximately $10^8$ viable spores) which were pregerminated in 500 μL of tryptic soy broth medium by incubation at 50 °C for 10 min. The mixture was spread onto MS medium. After incubation at 30 °C for 16 h, apramycin and nalidixic acid were added to the culture to the final concentrations of 50 μg/mL and 30 μg/mL, respectively. The transconjugants appeared after additional incubation for 2–5 days were isolated and streaked onto a fresh MS medium containing 10 μg/mL thiostrepton, 30 μg/mL nalidixic acid, and 2 mM theophylline. This induction culture was incubated at 30 °C for ~7 days. The colonies appeared after the cultivation was subjected to PCR and sequencing for selection of mutant. The selected mutant colonies were streaked onto MS medium without antibiotics and then the spores were collected. Plasmid loss was checked by spreading the spores onto apramycin-containing MS medium. For complementation of *cisS*, *cisS* and the promoter region upstream of *cisS* was amplified (primer set CisS_pTYM19t_Fw and Rv) and then fused with an integrative pTYM19t[46] plasmid digested with KpnI and HindIII. The resultant plasmid was transferred to *S. davawensis* as described above and transformants were selected by thiostrepton.

*S. davawensis* strains expressing the tagged spike tip protein was constructed as follows. *hrdB* promoter sequence and *BN159_7582* encoding spike tip protein of *S. davawensis* was amplified by PCR (primer set Tip_Fw, His6-GGGGSx2-Tip_Fw, and Tip_Rv) and then fused with the backbone region of pGMH plasmid[13] amplified using primer set pGMH_Inverse_Fw and Rv. A flexible linker (GGGGSGGGGS) was inserted between the hexahistidine tag and spike tip protein. The resultant plasmids (pGMHdw:: *his6-BN159_7582* and pGMHdw::*BN159_7582*) were transferred from *E. coli* to *S. davawensis* via conjugation as described above.

For heterologous expression of Tme and Tme-C, the expression plasmids were constructed as follows. *tme* was amplified by PCR (primer set Tme_Fw and Rv) and then fused with pET26b(+) digested with NdeI and HindIII. The codon-optimised Tme-C sequence was synthesised, amplified (primer set Tme-C_Fw and Rv), and fused with pET26b(+) as described above. Tme-C was designed to have an additional amino acid sequence (MNHKV) at the N-terminus which resulted in increased Tme-C yield. Site-directed mutation was introduced to the above plasmids by inverse PCR (primer sets H2051A_Fw and Rv; H23A_Fw and Rv; R21A_Fw and Rv; N37A_Fw and Rv; R51A_Fw and Rv; E55A_Fw and Rv) and subsequent phosphorylation/blunt end ligation. For heterologous expression of BN159_7588, *BN159_7588* sequence was amplified using a primer set BN159_7588_Fw and BN159_7588_Rv and the resultant fragment was fused with pET26b(+) as described above. For the heterologous expression of BN159_7587, *BN159_7587* was amplified using a primer set BN159_7587_Fw and BN159_7587_Rv and the resultant fragment was fused with pET26b(+) and 3×FLAG sequence.

For heterologous expression of Tme-N, the N-terminal region of Tme was amplified using a primer set Tme-N_Fw and Tme-N_Rv and the resultant fragment was fused with pColdII plasmid (Takara Bio Inc.; digested with BamHI and HindIII).

## Extraction of CIS particles

CIS particles were extracted from *S. davawensis* mycelia as follows. 50 μL of spore solution (approximately $10^5$ viable spores) were streaked onto MS medium using a disposable inoculation loop. After incubation at 30 °C for 3–4 days, the colonies were scraped off the plate and then lysed in the solution comprising 20 mM HEPES-NaOH (pH7.5), 150 mM NaCl, 1% (v/v) Triton X-100, 5 mg/mL egg white lysozyme, and a protease inhibitor cocktail. After incubation at 37 °C for 1.5 h, the solution was ultracentrifuged at $150,000 \times g$ for 1 h. The resultant pellet was resuspended in 500 μL of the resuspension buffer comprising 20 mM HEPES-NaOH (pH7.5), 150 mM NaCl, and 1 mM $MgCl_2$.

## Extraction and quantitative analyses of extracellular matrix

Extraction method for extracellular matrix of *S. davawensis* was based on a previous study.[47] The colonies were grown on a cellophane membrane placed onto MS medium. At each of the designated incubation time, the cellophane was peeled off and then soaked in 3 mL of 1.5 M NaCl solution for 5 min. The solution was transferred to a tube for centrifugation at $3000 \times g$ for 3 min to remove cells. The resultant supernatant was used for further analyses. The treated cellophane was not used for any experiment. DNA concentration in the extract was quantified using Quant-iT PicoGreen assay kit (Thermo Fisher, MA, USA). For agarose gel electrophoresis, 400 μL of the extract was subjected to ethanol precipitation and then dissolved in 40 μL of 10 mM HEPES-NaOH (pH7.5). Saccharide content was quantified by phenol-sulphate method. One aliquot of 5% phenol was added to the extract, and then 5 aliquots of sulphate was added. After cooling, absorbance at 490 nm was measured. Standard curve of glucose was used for calculation of the saccharide content in the extract. Protein concentration was quantified using Pierce BCA protein assay kit (Thermo Fischer). Standard curve of bovine serum albumin was used for calculation of the protein content in the extract.

## Microscopy

For transmission electron microscopy, samples were attached to thin carbon film-coated TEM grids (ALLIANCE Biosystems, Osaka, Japan) and washed with $H_2O$. The samples were then visualised by negative staining.

Reflection confocal images of *S. davawensis* colonies were acquired using an upright confocal microscope LSM880 (Carl Zeiss, Oberkochen, Germany) equipped with a Plan-Apochromat 10x objective lens (Carl Zeiss, 420640-9900, for Fig. 2e) or a LD C-Apochromat 40x objective lens (Carl Zeiss, 421867-9970, for Fig. 2f). In Fig. 2e, a 633 nm laser was irradiated, and the 623-641 nm reflected light was detected. In Fig. 2f, a 488 nm laser was irradiated, and the 472-490 nm reflected light was detected. Z-stacks were acquired at 3.11 μm intervals.

## Proteomic analysis

Spores of *S. davawensis* strains harbouring either pGMHdw:: *his6-BN159_7582* or pGMHdw::*BN159_7582* were streaked on MS medium containing 10 μg/mL thiostrepton and incubated at 30 °C for 2 days. The colonies were scraped off the plates and then disrupted by sonication in the buffer comprising Tris-HCl buffer (pH8.0) and 10 mM imidazole. After filtration (0.45 μm), the lysates were subjected to the $Ni^{2+}$ affinity chromatography using His GraviTrap column (Cytiva, MA, USA). The eluates were concentrated and buffer-exchanged with the above lysis buffer using Amicon Ultra-10K (Merck, Darmstadt, Germany).

Proteins were separated by SDS–PAGE and were treated by in-gel digestion with trypsin. The digested samples were purified by Zip-tips,

and were analysed by advance Nanoflor ultra-high performance liquid chromatography (Bruker, MA, USA) on a Q exactive quadrupole orbitrap mass spectrometer (Thermo Fisher) equipped with a Zaplous Column (0.2 i.d. × 50 mm; AMR, Inc., Japan, Tokyo) under the following conditions: column temperature, 35 °C; mobile phase, gradient mixture of solvent A [0.1% formic acid] and solvent B [acetonitrile]; flow rate, 1.5 mL/min; and gradient elution, 0 min (solvent A:solvent B = 95:5), 20 min (35:65) and 21 mn (5:95). For protein identification, quantification, and comparison between two groups, database search was performed by label-free quantification workflow in Proteome Discoverer 2.5 inserted with a Sequest HT search engine with percolator against the genome of *S. davawensis*. Abundances of peptide spectral matches were averaged from the two technical replicates. Abundance ratios and q-values were calculated from the results for three biological replicates.

## Protein expression and purification

*E. coli* BL21(DE3) cells harbouring pET26b::*tme-C-his$_6$*, pET26b::*tme-C(H23A)-his$_6$*, pET26b::*tme-C(R21A)-his$_6$*, pET26b::*tme-C(N37A)-his$_6$*, pET26b::*tme-C(R51A)-his$_6$*, pET26b::*tme-C(E55A)-his$_6$*, pET26b::*BN159_7588-his$_6$*, pET26b::*3×FLAG-BN159_7587*, or pCold::*tme-N* were precultured in liquid LB-Lennox medium containing 50 μg/mL kanamycin or ampicillin and then inoculated into 200 mL of the same medium. After incubation at 37 °C with shaking at 150 rpm, protein expression was induced by the addition of isopropyl-β-D-thiogalactopyranoside at the final concentration 0.05 (pCold::*tme-N*), 0.1 (pET26b::*tme-C-his$_6$*, pET26b::*tme-C(H23A)-his$_6$*, pET26b::*tme-C(R21A)-his$_6$*, pET26b::*tme-C(N37A)-his$_6$*, pET26b::*tme-C(R51A)-his$_6$*, pET26b::*tme-C(E55A)-his$_6$*, and pET26b::*BN159_7588-his$_6$*), or 0.25 (pET26b::*3×FLAG-BN159_7587*) mM. The culture was further incubated at 18 °C for 18 h. The cells were harvested by centrifugation and then disrupted by sonication in 5 mL of the lysis buffer comprising Tris-HCl buffer (pH 8.0) and 10 mM imidazole. For purification, the filtered lysates were subjected to the Ni$^{2+}$ affinity chromatography using His GraviTrap column. The eluates were concentrated and buffer-exchanged with 10 mM HEPES-NaOH (pH 7.5) using Amicon Ultra-10K. For an interaction assay using BN159_7588-His$_6$ and 3×FLAG-BN159_7587, the lysates (1 mL per strain) were mixed and then incubated for 20 min at room temperature. The mixture was subjected to the Ni$^{2+}$ affinity chromatography as described above. The eluates were directly subjected to western blotting.

## Enzyme assays

The nucleolytic activity of Tme-C was evaluated under the following conditions. Standard reaction mixture was comprised of 50 μg/mL Tme-C, 20 mM HEPES-NaOH (pH7.5), and the shorter (double-stranded pUC19 and single-stranded M13mp18 virion DNA; 2.5 ng/μL; extracted total RNA, 10 ng/μL) or the longer substrate (genomic DNA; 50 or 100 ng/μL). The reaction mixture was incubated at 30 °C, and then the reaction was stopped by cooling on ice. All reaction mixtures were mixed with 0.05% (w/v) SDS and then subjected to agarose gel electrophoresis. Polynucleotides were visualised with ethidium bromide.

## Bioinformatic analyses

The amino acid sequences of the CIS tube and sheath proteins were obtained from database for eCIS[5] and listed in Supplementary Tables 3 and 4. Evolutionary analyses were conducted in MEGA X.[48] All ambiguous positions were removed from each sequence pair (pairwise deletion option). Structures of CIS proteins were modelled using ColabFold ver. 1.5.2[49] with multiple sequence alignment (msa_mode, mmseq2_uniref_env; pair_mode, unpaired_paired). Domain search and homology search were performed by NCBI conserved domains database[50] and HHpred,[51] respectively. CPP prediction was performed by C2Pred.[31]

## Western blotting

Rabbit antisera against Tme-C and Tme-N were developed using an internal peptide of the C-terminal (DWREKGETKNWSQDPDPIA) and N-terminal (RRRKRKERAAKSRTPEPKN) regions, respectively, as antigens. Proteins were separated by SDS–PAGE with a 4-15% Mini-PROTEAN TGX Gel (Bio-Rad Laboratories, Inc., CA, USA) and then electroblotted onto a PVDF membrane. After blocking with 5% (w/v) skim milk in Tris-buffered saline supplemented with 0.02% (v/v) Tween 20 (TBS-T buffer), the blots were incubated with each antibody (anti-Tme-C serum, diluted to 0.1%; anti-Tme-N serum, diluted to 0.05%; anti-His$_6$ antibody AB9108 [Abcam, Cambridge, UK], diluted to 0.02%; anti-DDDDK tag antibody AB1162 [Abcam], diluted to 0.01%) at room temperature for 60 min and horseradish peroxidase-conjugated secondary antibody (Goat anti-rabbit IgG H&L [HRP] ab6721[Abcam], diluted to 0.01%) at room temperature for 45 min. Transblotting was performed using Tris/glycine system with (10% [v/v]; anti-Tme-C-serum, anti-His$_6$ antibody, anti-DDDDK antibody; anti-Tme-N serum for the truncated Tme-N) or without (anti-Tme-N serum for Tme and its variant) methanol. All the primary antibodies were diluted with the blocking buffer. Secondary antibody was diluted with TBS-T buffer (for anti-Tme-C-serum, anti-His$_6$ antibody, anti-DDDDK antibody) or the blocking buffer (for anti-Tme-N serum). The immunoreactive proteins were detected with ImmunoStar LD (FUJIFILM Wako Chemicals, Osaka, Japan). After adding the substrate solution, chemiluminescence signals were detected immediately (anti-Tme-C-serum, anti-His$_6$ antibody, anti-DDDDK antibody, anti-Tme-N serum for the truncated Tme-N) or in 15 min (anti-Tme-N serum for Tme and its variant).

## Reporting summary

Further information on research design is available in the Nature Portfolio Reporting Summary linked to this article.

## Data availability

Source data are provided with this paper. The predicted protein structures are provided in figshare (https://doi.org/10.6084/m9.figshare.25699534.v1). The proteomic raw data are available in jPOSTrepo (JPST003078; PXD052139). The previously reported structure of EndA is available in the Protein Data Bank (3OWV). The previously reported genome sequence of *S. davawensis* JCM 4913 is available in the GenBank database (HE971709.1). Any other datasets generated for the current study are available from the corresponding authors on request. Source data are provided with this paper.

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

## Acknowledgements

T.Nagakubo was supported by a Grant-in-Aid for Scientific Research from the Japanese Society for the Promotion of Science (19K15726 and 23K13863) and Institute for Fermentation (Osaka, Japan). M. T. was supported by a Grant-in-Aid for Scientific Research from the Japanese Society for the Promotion of Science (23K26811) and the Suntory Rising Stars Encouragement Program in Life Sciences (SunRiSE). N. N. was supported by the Japan Science and Technology Agency (JPMJMI21G8 and JPMJGX23B2). We thank Dr. Shiori Doi (Keio University, Japan) for technical assistance. We also thank Prof. Hiroyasu Onaka (Gakushuin University, Japan) for providing pTYM19t plasmid.

## Author contributions

T.Nagakubo designed the study. T.Nagakubo and T.Y. performed the microscopic analyses. T.Nishiyama performed proteomic analysis. T.Nagakubo performed all other experiments. T.Nagakubo and T.Nishiyama analysed the data. T.Nagakubo and T.Y. drafted the manuscript. T.Nagakubo, T.Nishiyama, T.Y., N.N., and M.T. discussed the results and commented on the manuscript.

## Competing interests

The authors declare no competing interests.
