## [Peer Review File · Nature Communications]

Contractile injection systems facilitate sporogenic differentiation of *Streptomyces davawensis* through the action of a phage tapemeasure protein-related effector.Reviewer #1 (Remarks to the Author):

The manuscript "Contractile injection systems facilitate sporogenic differentiation of bacteria through the action of a phage tapemeasure protein-related effector" shows the role of a contractile injection system (CIS) in aerial mycelium differentiation and sporulation in *Streptomyces davawensis* dense cultures. The authors show that *Streptomyces davawensis* CIS proteins have homologies with diverse bacterial phyla and are distant to the other well-studied actinobacterial CISs and SLPs as those from *Streptomyces coelicolor* and *Streptomyces lividans*; authors also demonstrate that the CIS proteins that they study are involved in modulate extracellular matrix composition (eDNA and saccharides); they identified a DUF4157 protein that is putatively ejected by *S. davawensis* CISs; this DUF4157 protein was characterised as a tapemeasure protein-related endonuclease and named as "tapemeasure-related effector" (Tme); the Tme endonuclease activity was associated with the CIS-associated phenotypes of *S. davawensis*.

The manuscript is well written, the results well-presented and to the best of my knowledge the results are novel.

Some points to be considered prior publication:

MAJOR:

- Is difficult to discern the importance of the CIS proteins and tapemeasure endonuclease (Tme) characterised in this work in *Streptomyces* biology. If the CIS proteins and Tme analysed in this work are not conserved in the *Streptomyces* genus, is difficult to think in a key role of these proteins in *Streptomyces* differentiation. Can authors make a comparison with other *Streptomyces* and discuss this aspect? All the *Streptomyces* have CIS proteins? Are there CIS proteins conserved in *Streptomyces*? If not, are the CIS proteins from each *Streptomyces* making similar functions in differentiation?

- Is the DUF4157 tapemeasure endonuclease conserved in *Streptomyces*?

- It's difficult to understand the exact CIS protein/s that are being analysed in the different sections of this work. Fig. 1A shows a gene cluster encoding CIS structural proteins.

- Are all the proteins encoded in this cluster, CIS proteins? Please clarify.

- Fig. 1B shows a phylogenetic tree of the CIS tube proteins. What happens with the rest of the proteins encoded in the cluster? Can authors make a phylogenetic tree for these proteins, or discuss why they only make the tree for the CIS tube protein?

- Fig. 1B what genes/proteins are used in the tree? Can authors indicate the genes/proteins accession numbers?

- Fig. 1C. CRISPR-Cas9 of the *cisS* gene. Why mutate *cisS* and not the other genes?

- It looks cryptic why authors study *Streptomyces davawensis*. Made the authors a screening for uncharacterised CIS proteins in *Streptomyces* and found this species? Or authors were already working with this *Streptomyces* species?

- Pages 8 and 9, endonuclease activity. I do not fully understand this important part of the manuscript:

- Fig. 4a Tme has 2047aa; why in Fig. 4b Tme has 2076aa?

- I guess that Tme-C means the carboxyl part of the Tme protein. Please clarify exactly the aminoacids contained in Tme-C. Which part of the protein was exactly eliminated in TmeC? Why authors know that this part of the protein can be important for the nuclease activity?

- Native Tme is toxic for *E. coli*; but histidine 2051 substitution by Ala allows Tme overexpression in *E. coli*; Native TmeC seems to be less toxic. Why?. Please, label and specify in Fig. 4D that these results are from TmeC; and in Fig. 4C that results are from Tme

- Why TmeC can be expressed (and purified) in *E. coli* if it still having nuclease activity? Is this nuclease activity lesser than the nuclease activity of the native Tme? I do not see pUC19 degradation by TmeC in Fig. 4F; why the difference in the pUC19 mobility in Fig. 4F? Is it consequence of the TmeC binding to the linear pUC19? I am familiar with the analysis of interaction of proteins/compounds with circular plasmids in agarose gels: interaction with circular plasmidic DNA changes the ratio between open and supercoiled plasmid forms. However, in this manuscript it seems that the linear pUC19 plasmid was used (I only see a band in Fig. 4F and it is

labelled as linear pUC19). To detect this heavy change the mobility of a linear plasmid mobility in an agarose gel, TmeC should be a very big protein. Please, clarify this result.

- Supplementary Figure 10F I do not understand why at 2-hours circular DNA appears
- If Fig. 4F and Supplementary Figure 10 are agarose gels, please, indicate it in the figure legends and in the manuscript. If they are not agarose gels, please, indicate what are they.

MINOR:

- Title: as all work focus in *Streptomyces davawensis* I would change "sporogenic differentiation of bacteria" by "sporogenic differentiation of *STREPTOMYCES DAVAWENSIS*". I would include both, genus and species, because it seems that the CIS studied in this work is characteristic of *S. davawensis* and is not present in other *Streptomyces* (Fig. 1a)

- page 4 line 15: "biological function remains limited to only a few bacterial species" Please, include some references

- page 5, Line 10. Please, define the meaning of SLP (I guess "Streptomyces phage tail-like particles").

- page 5, line 24 "extremely low efficiency of homologous recombination at the genomic locus" How authors know that there is a low efficiency of homologous recombination at the genomic locus? Is this a hypothesis? Homologous recombination was quantified? Please, explain.

- Fig. 2A y axis. Biomass refers to dry weight or fresh weight? Despite that differences in weight are statistically significant, they look very low. In addition to biomass weight, authors can also quantify total protein per plate (in the mycelium collected from cellophane).

Reviewer #2 (Remarks to the Author):

The manuscript under consideration delves into the realm of Contractile Injection Systems (CISs), intricate prokaryotic nanostructures akin to phage tails that harbor effector proteins orchestrating diverse biological processes. While acknowledging the evolutionary diversification of CIS functions and their potential as promising protein delivery systems, the current functional characterization of CISs and their effectors is notably confined to a limited spectrum of CIS lineages. The presented research sheds light on the unique attributes of CISs within *Streptomyces davawensis*, classifying them as a distinctive subset of bacterial CISs distributed across distant phyla and integral to the facilitation of sporogenic differentiation in this bacterium. Loss of CISs is correlated with observable reductions in extracellular DNA release, biomass accumulation, and spore formation in *S. davawensis*. The study further delineates the effector loaded onto CISs, identifying it as a remote homolog of phage tailmeasure proteins, with its C-terminal domain housing endonuclease activity accountable for the discernible CIS-associated phenotypes. The findings presented herein not only underscore the vital role of CISs in bacterial reproduction through effector action but also propose an intriguing evolutionary nexus between CIS effectors and viral cargos.

The clarity of the manuscript is commendable, offering valuable insights into the regulatory role of CIS systems in *Streptomyces* development. Notably, it unveils, for the first time in these organisms, an effector associated with the CIS system. I have a few inquiries and suggestions aimed at enhancing clarity on certain aspects that remain to be fully elucidated.

1) p.5 lines 6-17: The author conducted a phylogenetic analysis employing the CIS tube protein to assert the uniqueness of the *S. davawensis* CIS group. However, it's worth noting that CIS gene clusters typically encompass at least two tube proteins rather than a singular one. Is there experimental evidence validating the significance of this specific tube protein in the formation of CIS particles? In contrast, the manuscript and other articles robustly demonstrate the significance of the sheath protein. Utilizing the sheath protein for phylogenetic analysis would likely offer a

more compelling argument.

2) p.5, line 16: The authors illustrate the uniqueness of *S. davawensis* CIS. It would be informative to elucidate the structural and elemental similarities or differences between *S. davawensis* CIS and well-known counterparts like *S. coelicolor* or *S. venezuela*. A schematic illustration of *S. davawensis* CIS could prove beneficial in aiding readers' comprehension.

3) The authors generated a mutant by introducing a stop codon at Gln78 of BN159_7569 (CisS), effectively preventing the translation of the sheath protein in CisS. However, in line 6 on page 6, it is stated that the granule structure consists of CISs composed of baseplate protein. How do the authors ensure that the translation of other genes downstream of CisS are not prematurely terminated? Have the author conducted analyses to detect the expression of other genes within this gene cluster?

4) While I observe consistency between the trends asserted by the authors, it's noteworthy that the P-values in Figure 2 (G-I) do not consistently reach a "significant" threshold, thereby casting doubt on the validity of certain conclusions drawn. A similar observation can be made for Fig. 4K (P=0.068).

5) Have the authors investigated Programmed Cell Death (PCD) in connection to their CIS system, such as through the application of live/dead staining, a method previously employed in the study of other CIS systems in *Streptomyces*?

6) In lines 17-19 on page 6, the authors show that the mutant exhibits impaired multicellular development, specifically in aerial hyphae and spore formation. Is there a potential correlation between these morphological changes and the production of secondary metabolites? Has the author conducted a comparison of resorflavin production between the wild type (WT) and the mutant to explore this aspect further?

7) Protein interactions play a significant role in supporting several conclusions made by the authors, as exemplified in lines 28-36 on page 7. The author utilized a pull-down assay to identify partners of the CIS tip protein. Were alternative methods employed to directly validate the interaction between BN159_7576 and the tip protein? Furthermore, in lines 6-8 on page 10, the authors employed AlphaFold to predict protein interactions between BN159_7588 and BN159_7587. However, given the artificial nature of AlphaFold predictions, additional evidence, such as bacterial 2-hybrid assays, should be provided by the authors to strengthen their claims.

8) The assertion in line 19 on page 10, suggesting that CIS particles are unaffected, lacks convincing support. Figure 2H is inadequately detailed, making it challenging to draw a meaningful comparison between the CIS particles of the *tme* Q2006 mutant and the wild type for a conclusive determination.

9) In lines 10-12 on page 10, the pivotal conclusion regarding the ability of Tme to degrade genomic DNA requires additional substantiation. The author introduced the Tme-C H23A point mutation, but the evaluation was limited to the nucleolytic activity on the PUC19 plasmid. Why did the author exclusively examine this plasmid and not include genomic DNA from *S. davawensis*? Is there a possibility that this point mutation may not effectively impede the nucleolytic activity of Tme-C on *S. davawensis* genomic DNA? Furthermore, has the author explored whether Tme-C can degrade RNA from *S. davawensis*? Additional experimentation is needed to address these aspects comprehensively.

10) Does the expression pattern of Tme align with that of the CIS system during the development of *S. davawensis*? If not, what mechanisms does *S. davawensis* employ to safeguard itself from genome degradation by Tme?

Reviewer #3 (Remarks to the Author):

In this manuscript, Nagakubo and colleagues study a prototype of a new group of contractile injection systems distributed in diverse phyla. The role of this CIS is clearly important for the biology of *Streptomyces davawensis*, although this function needs to be understood in the future. Furthermore, the putative cognate effector is distantly related to phage tail measure proteins and bears endonuclease activity. Hence, the molecular mechanism supporting this CIS functions is also quite interesting, although again, the manuscript does not provide a clear hypothesis. Finally, the implementation of a CRISPR-Cas9N-based editing system is very useful here.

I really liked reading the paper. However, I have several problems with the data. As it is, I don't think the conclusions proposed by the authors are totally supported by the data and require additional explanation or results.

I also think the discussion could be enhanced. Why do the authors exclude the possibility that Tme-dependent cell lysis (since no immunity is there?) could result in DNA release in the extracellular environment, instead of resulting from a pore-forming and DNA cleavage activity in targeted cells? Again, I may have missed a point here. Also, why don't the authors discuss the presence or absence of an immunity protein? It may well not be necessary in this particular system, but it would be nice to explain why.

Specific major issues I have:

Section starting P6 L22:

How was biomass measured? I am not convinced by the measurements showing a decreased growth in phase 1, as other differences mentioned could possibly affect these measurements. In addition, although it is statistically significant, it appears minimal and only shown in one measure. Such claim should be backed up with more points. Anyway, is it relevant at all?

Fig. 4G: the results do not clearly show if the Tme Q2006* protein is produced at all. Hence, the authors cannot specifically conclude that Tme nuclease activity is responsible for the phenotypes observed (even though I am totally convinced Tme is required).

Sup Fig. 5; I am having a hard time believing how the authors interpret the results or the result itself. First, the picture may have been contrasted excessively, therefore hiding the smear gDNA should show. Yet, wouldn't we expect Tme activity to results is a more pronounced DNA smear. Here it would be as if gDNA is cut at specific sites.

Other specific comments:

L10: wording is ambiguous. Are all three processes decreased or only DNA release, while biomass accumulation and spore formation are promoted.

L13: it would be more interesting to discuss the diversity of molecular activities associated with CISs so far rather than just vaguely mentioning "few bacterial species"

P3, paragraph starting L28, more specifically P4 Lines 4: The way "host" is used is kind of confusing.

P6 L1: in which conditions were the lysates prepared?

P6 L15: Fig 1E-F and not 1D-F?

P6, Paragraph L10: it is not clear to me why grow rate was not compared first, and only mentioned in the next paragraph.

Figure 4F: can the authors better explain why there is a light upper band when pUC19 was incubated with Tme-C please? Do they mean that the His23Ala mutant cannot bind DNA? Is it known if this residue is required for DNA binding?

Also, the Materials and Methods section does not give precision, but I am amazed that Tme was so easily produced and purified. Yet, Fig 4C clearly shows cytotoxicity of Tme produced in BL21 cells.

Maybe I missed a particular detail.

P12 L1: Can the authors provide any data to support CPP-like properties? Did they use CPP detection tools available? The whole Tme-pore hypothesis seems a bit bold. The pore may be provided by a receptor protein interacting with Tme.

To Reviewer #1:

We greatly appreciate the time and the insightful comments on our manuscript. We are particularly grateful to the reviewer for the recognition of the manuscript's clarity and the novelty of our findings, and kindly mentioning “*The manuscript is well written, the results well-presented and to the best of my knowledge the results are novel*”. We have carefully considered all the comments and have thoroughly addressed all the feedback provided in the following section. Along with these, we have also made specific revisions to the manuscript and figures by refining the text or adding data to ensure clarity and readers' comprehensiveness. The changes on the manuscript are highlighted in blue. Furthermore, we have expanded specific discussions to Supplementary Notes to provide deeper context and interpretation of the findings without exceeding the content of the main text. Point-by-point responses to the comments are as follows.

MAJOR:

- Is difficult to discern the importance of the CIS proteins and tapemeasure endonuclease (Tme) characterised in this work in *Streptomyces* biology. If the CIS proteins and Tme analysed in this work are not conserved in the *Streptomyces* genus, is difficult to think in a key role of these proteins in *Streptomyces* differentiation. Can authors make a comparison with other *Streptomyces* and discuss this aspect? All the *Streptomyces* have CIS proteins? Are there CIS proteins conserved in *Streptomyces*? If not, are the CIS proteins from each *Streptomyces* making similar functions in differentiation?

We interpret the biological significance of the *Streptomyces* CISs as supportive but not essential systems facilitating the morphological differentiation of producer bacteria under certain growth conditions. Previous studies have shown that CISs are conserved in an estimated ~90% of *Streptomyces* species (Chen *et al. Cell Reports* 2019). As shown in previous studies conducted by us and other groups, most of *Streptomyces* CISs belong to an actinomycetes-exclusive CIS lineage (Nagakubo *et al. mSphere* 2023; Casu *et al. Nature Microbiology* 2023; Vladimirov *et al. Nature Communications* 2023), suggesting that these typical actinomycetes CISs have propagated vertically during divergence of actinomycete species from a common ancestor. Although the detailed functions remain largely elusive, the previous studies have consistently

demonstrated that the typical actinomycetes CISs are not essential for the morphological differentiation at least in *S. lividans* and *S. coelicolor* (Nagakubo *et al. mSphere* 2023; Casu *et al. Nature Microbiology* 2023; Vladimirov *et al. Nature Communications* 2023). Rather, this type of CISs is likely to confer ecological benefits to the producer bacteria by indirectly affecting the differentiation under certain ecological conditions. This would imply that, despite their high conservation among *Streptomyces*, CISs have been employed by these bacteria as non-essential systems which indirectly contribute to their life cycle and can potentially be replaced with other systems.

It is relatively rare that multiple CIS gene clusters exist in a single *Streptomyces* genome, possibly due to potential promiscuity in subunit-subunit interactions among different types of CISs resulting in misassembly of protein complexes. Therefore, the non-essentiality of the “typical” actinomycetes CIS would allow for the replacement with “unique” CISs, such as the Tme-harboursing CIS analysed in the current study, in a *Streptomyces* species when both CISs affect the morphological differentiation, and the latter CIS is more advantageous to the bacterium. Ecological niches each *Streptomyces* species has established, and the niche-associated selective pressures may determine which types of CISs are more advantageous and ultimately drive the selection of CISs. In this viewpoint, the ecological background of *S. davawensis* originally isolated from a tropical soil sample might be related to the acquisition of the novel type of CISs.

Finally, we believe that the function mechanisms would be different between the *S. coelicolor/S. lividans* CISs and the *S. davawensis* CISs for the following reasons:

(i) Difference in the effector proteins

Tme-like effector, which is defined by the distinct modular architecture, genomic synteny, and homology with phage tapemeasure proteins, is not encoded in the *S. coelicolor/S. lividans* CIS clusters. Casu *et al.* proposed ricin-like proteins as effectors of the *S. coelicolor* CIS. Although we could not detect such proteins from the counterpart in *S. lividans*, we have detected another putative effector protein from the *S. lividans* CIS, and this effector protein is likely to have different activity from that of Tme of *S. davawensis*. The identification of the putative effector protein in the *S. lividans* CIS is not reported anywhere and the manuscript including these data will soon be submitted to a scientific journal by us.

(ii) Difference in the CIS-associated phenotypes

As presented in the current manuscript, the *S. davawensis* CISs increase the biomass, eDNA release, and spore formation of the bacterium in dense culture. On the other hand, there is no clear evidence that the *S. coelicolor*/*S. lividans* CISs increase the biomass. Regarding spore formation, the phenotypic consequence may differ depending on the species. The authors claimed that CISs of *S. coelicolor* mediate the cell death and therefore delay spore formation (Casu *et al. Nature Microbiology* 2023). In *S. lividans*, loss of CISs (referred to as SLPs in the paper) resulted in decreased spore formation in high osmolarity medium (Nagakubo *et al. mSphere* 2023). For comparison, in prior to the initial submission, we cultivated *S. davawensis* in the above high osmolarity medium used for *S. lividans* but there was no significant difference in *S. davawensis* phenotypes with or without CISs. eDNA release was not reported for *S. coelicolor*, but the CIS loss rather increased eDNA release in *S. lividans* under high osmolarity condition, probably due to the higher sensitivity to the osmotic stress. Therefore, we believe that the CISs with different functionalities caused the different phenotypic consequences.

We added the above discussions to the sections 2. *Comparison of Streptomyces CIS gene clusters* and 3. *Biological significance of CISs among Streptomyces species* in Supplementary Notes. Related description was also added to the Discussion (page 13, lines 33-36). We excluded several points, including unpublished results and hypotheses requiring further verification, from the additional discussions.

- Is the DUF4157 tapemeasure endonuclease conserved in Streptomyces?

On a BLAST search in the NCBI database, DUF4157 domain-containing tapemeasure protein-related effector (Tme) homologs, containing consensus endonuclease motifs, can be found in CIS gene clusters of at least 47 of the *Streptomyces* genomes including *S. griseus*. These Tme homologs may have similar function to the *S. davawensis* Tme. For reference, we attached a part of the BLAST search result where the Tme polypeptide from *S. davawensis* JCM 4913 was subjected as Query. Red lines indicate homologous regions of the hit proteins with high amino acid sequence similarity to Tme. All these hits are encoded within the *Streptomyces* genomes. Interestingly, in some *Streptomyces* species, the C-terminal part of Tme seems to be replaced with other domains with unknown function whereas the other part is relatively highly conserved. This would suggest that these species have the Tme homologs with different C-terminal activities, highlighting the diverse strategies of *Streptomyces* to employ the phage tapemeasure protein-related CIS effectors. Therefore, the discovery of Tme and its homologs would open a new avenue to understand the intriguing ability of *Streptomyces* species to coexist with and utilise the CIS effectors like Tme, probably originating from viral infection systems.

- It's difficult to understand the exact CIS protein/s that are being analysed in the different sections of this work. Fig. 1A shows a gene cluster encoding CIS structural proteins.

Thank you for the comment. We added a part of the CIS gene cluster to Fig. 3b where several CIS proteins appear. We believe that this would help readers understand which CIS proteins are mentioned in this section.

- Are all the proteins encoded in this cluster, CIS proteins? Please clarify.

Annotations of several proteins were inferred from homology with CIS structural proteins or experimental results in the current study. As for the other proteins encoded in the gene cluster, we are currently unable to assign their structural/functional roles due to the lack of evidence. Some details are provided in the section 2. *Comparison of Streptomyces CIS gene clusters* in Supplementary Notes.

- Fig. 1B shows a phylogenetic tree of the CIS tube proteins. What happens with the rest of the proteins encoded in the cluster? Can authors make a phylogenetic tree for these proteins, or discuss why they only make the tree for the CIS tube protein?

We conducted a phylogenetic analysis for the CIS tube proteins because the amino acid sequences of tube proteins are relatively highly conserved compared with other structural proteins and their functional and evolutionary significance has already been demonstrated in other CISs (Shikuma *et al. Science* 2014; Jiang *et al. Cell* 2019). According to the reviewers' suggestion, we constructed a tree for CIS sheath proteins (Supplementary Fig. 1). Overall, this tree is consistent with the tree for tube proteins with respect to the evolutionary uniqueness of the *S. davawensis* CIS. This additional analysis employing the sheath proteins was also recommended by the reviewer #2.

- Fig. 1B what genes/proteins are used in the tree? Can authors indicate the genes/proteins accession numbers?

We provide the accession numbers of CIS proteins used in the phylogenetic analysis in Supplementary Table 3. According to the addition of the phylogenetic tree for sheath proteins, they were also included in the manuscript as Supplementary Table 4.

- Fig. 1C. CRISPR-Cas9 of the *cisS* gene. Why mutate *cisS* and not the other genes?

We mutated *cisS* encoding the sheath protein for the following reasons:

(i) Deletion or mutation of sheath protein(s) is an effective and general approach to abolish CIS functions (Nagakubo *et al. mSphere* 2023; Casu *et al. Nature Microbiology* 2023).

(ii) Although a mutation of a spike complex can be another approach to abolish CIS functions, the potential redundancy of spike protein-encoding genes (*Afp7* and *Afp8* homologs) of the *S. davawensis* CIS cluster was suggested from the homology analysis performed by Chen *et al. (Cell Reports* 2019; http://www.mgc.ac.cn/cgi-bin/dbeCIS/showecis.cgi?id=GCA_000349325.1). To avoid multiple genome editing potentially leading to the accumulation of unintended mutations, we targeted the sheath protein encoded as a singular gene *cisS*.

- It looks cryptic why authors study *Streptomyces davawensis*. Made the authors a screening for uncharacterised CIS proteins in *Streptomyces* and found this species? Or authors were already working with this *Streptomyces* species?

We have conducted neither a screening nor other works on *S. davawensis*. In our previous study, we found that the CIS-related nanoparticle produced by *S. lividans* (SLP) represents a distinct CIS lineage encompassing CISs conserved exclusively among actinomycetes including the genus *Streptomyces* (Nagakubo *et al. mSphere* 2023). During the research, we also noticed that some *Streptomyces* species have CIS gene clusters which are not similar to the above “typical” actinomycetes CIS gene clusters with respect to the amino acid sequences of structural proteins and the synteny of gene clusters. We were interested in why such non-typical CISs have been selected by some *Streptomyces* species through evolution. We then hypothesised that they have lost the “typical” actinomycetes CISs and instead acquired the “non-typical” CISs that provide considerable benefits to them (please see also the section 3. *Biological significance of CISs among Streptomyces* in Supplementary Notes). To expand our research to the new area of bacterial CISs, we have investigated *Streptomyces* species having the “non-typical” CISs, and one of them is *S. davawensis* JCM4913 readily available from RIKEN BRC, a national bioresource center in Japan.

- Pages 8 and 9, endonuclease activity. I do not fully understand this important part of the manuscript:

- Fig. 4a Tme has 2047aa; why in Fig. 4b Tme has 2076aa?

Total length of the Tme polypeptide is 2247 amino acids (Fig. 4a). Fig. 4b shows the partial amino acid sequence ranging 2047-2076 of the Tme polypeptide.

- I guess that Tme-C means the carboxyl part of the Tme protein. Please clarify exactly the aminoacids contained in Tme-C. Which part of the protein was exactly eliminated in TmeC? Why authors know that this part of the protein can be important for the nuclease activity?

As described in the Fig. 4A legend of the initially submitted manuscript, Tme-C contains amino acids 2030-2247 of Tme. For more clarity, we modified the manuscript to emphasise this point (page 9, line 3).

As described in the initially submitted manuscript, the C-terminal part of Tme was predicted as an endonuclease-like domain by HHpred. This finding led us to further identify key endonuclease motifs within the amino acid sequence of Tme. Also, the AlphaFold2 structure of this domain showed the similar conformation of essential active site amino acids, supporting the above prediction (Supplementary Fig. 10).

- Native Tme is toxic for *E. coli*; but histidine 2051 substitution by Ala allows Tme overexpression in *E. coli*; Native TmeC seems to be less toxic. Why?. Please, label and specify in Fig. 4D that these results are from TmeC; and in Fig. 4C that results are from Tme

As the reviewer pointed out, the native Tme-C seems less toxic to *E. coli* than the full-length Tme. Although we currently cannot provide the detailed mechanism of this phenomenon, the N-terminal DUF4157 domain, a proposed core domain of many CIS effectors, and the hydrophobic regions may be implicated in the toxicity of Tme. To note, the importance of DUF4157 domain in activities of CIS/T6SS effectors has been proposed in previous studies (Geller *et al. Nature Communications* 2021; Wood *et al. Cell Reports* 2019). Given that DUF4157 domain potentially interacts with bacterial cell envelope-associated proteins (Wood *et al. Cell Reports* 2019), the corresponding N-terminal domain of Tme might enhance the toxicity of the C-terminal nucleolytic activity through interactions with cell envelope-associated protein(s). Alternatively, the hydrophobic regions might enhance the toxicity. For better interpretation and

presentation of the results, we added the related sentence and labels to the manuscript (page 9, lines 3-5).

- Why TmeC can be expressed (and purified) in *E. coli* if it still having nuclease activity? Is this nuclease activity lesser than the nuclease activity of the native Tme? I do not see pUC19 degradation by TmeC in Fig. 4F; why the difference in the pUC19 mobility in Fig. 4F? Is it consequence of the TmeC binding to the linear pUC19? I am familiar with the analysis of interaction of proteins/compounds with circular plasmids in agarose gels: interaction with circular plasmidic DNA changes the ratio between open and supercoiled plasmid forms. However, in this manuscript it seems that the linear pUC19 plasmid was used (I only see a band in Fig. 4F and it is labelled as linear pUC19). To detect this heavy change the mobility of a linear plasmid mobility in an agarose gel, TmeC should be a very big protein. Please, clarify this result.

We believe that the nucleolytic activity and cytotoxicity of Tme-C is very weak compared with other EndA-related endonucleases and therefore we could have isolated this enzymatic domain from the *E. coli* cells. The band of 2.5 ng/ μ L of pUC19 disappeared after incubation with 50 ng/ μ L of Tme-C at 30°C for 180 min, suggesting that the Tme-C-catalysed hydrolysis is very slow compared with EndA (please see also the section 4. *Unique enzymatic characteristics of Tme-C* in Supplementary Notes). Please note that the cytotoxicity caused by Tme/Tme-C was observed when these proteins are overexpressed with the extremely strong T7 RNA polymerase/T7 promoter system in *E. coli*. Although we cannot simply compare the nucleolytic activity of Tme-C and the full-length Tme, the N-terminal region of Tme would contribute to the toxic effect (please see the response to the above comment). We are currently unable to demonstrate whether the N-terminal region directly affects the activity of Tme-C or not. Still, some of Tme-C expressing *E. coli* cells could survive and grow slowly probably due to relatively low nucleolytic activity of Tme-C.

We have several data implying that the Tme-C activity is limited to a low level. Details are described in the section 4. *Unique enzymatic characteristics of Tme-C* in Supplementary Notes. In brief, Tme appears to lack several amino acid residues that facilitate hydrolysis of the DNA phosphodiester in *S. pneumoniae* EndA. In addition, Tme-C can form a stable, probable covalent complex with dsDNA substrate, which could be a rate-limiting step in the Tme-C-catalysed nucleolytic reaction and lead to the

upward shift of the DNA band in agarose gel. Regarding this, we provide the additional experimental results in Supplementary Fig. 13. According to the addition of these data, we modified and expanded the related descriptions in the Results section (page 9, lines 6-36; page 10, line 1).

- Supplementary Figure 10F I do not understand why at 2-hours circular DNA appears

We apologise for the misleading presentation of the data. In this panel (Supplementary Fig. 12d in the revised manuscript), “M13mp8 circular single-stranded DNA” indicates the substrate for the Tme-C-catalysed reaction. To avoid misleading, we modified the panel and clarified that the circular single-stranded DNA was used as a substrate. Although we did not identify the weak band, which the reviewer has mentioned, we speculate that this would be a linear degradation product derived from M13mp8 circular single-stranded DNA. As we discussed in Supplementary Notes, our data suggest the differential recognition of dsDNA/ssDNA substrates by Tme-C. Notably, Tme-C seems not to form a probable covalent complex with ssDNA, and this may be due to distinct substrate conformations approaching to the catalytic site (please see also the final section in Supplementary Notes). The absence of pairing nucleotides in ssDNA might make Tme-C-substrate interaction less efficient at a certain site(s) of the ssDNA substrate, possibly resulting in the appearance of the weak band below the substrate circular ssDNA.

- If Fig. 4F and Supplementary Figure 10 are agarose gels, please, indicate it in the figure legends and in the manuscript. If they are not agarose gels, please, indicate what are they.

According to the suggestion from the reviewer, we indicated gel types in the figure legends.

MINOR:

- Title: as all work focus in *Streptomyces davawensis* I would change “sporogenic differentiation of bacteria” by “sporogenic differentiation of STREPTOMYCES DAVAWENSIS”. I would include both, genus and species, because it seems that the

CIS studied in this work is characteristic of *S. davawensis* and is not present in other *Streptomyces* (Fig. 1a)

Thank you for the suggestion. We modified the title to specify the research subject.

- page 4 line 15: “biological function remains limited to only a few bacterial species”
Please, include some references

According to the reviewers' suggestions, we added several sentences and references to the end of the paragraph (page 3, lines 17-24).

- page 5, Line 10. Please, define the meaning of SLP (I guess “*Streptomyces* phage tail-like particles”).

According to the suggestion from the reviewer, we added the definition of SLP (page 5, line 11).

- page 5, line 24 “extremely low efficiency of homologous recombination at the genomic locus” How authors know that there is a low efficiency of homologous recombination at the genomic locus? Is this a hypothesis? Homologous recombination was quantified? Please, explain.

It is a hypothesis as we did not quantify homologous recombination at the locus. We modified the related sentence to clarify this point (page 5, line 25).

- Fig. 2A y axis. Biomass refers to dry weight or fresh weight? Despite that differences in weight are statistically significant, they look very low. In addition to biomass weight, authors can also quantify total protein per plate (in the mycelium collected from cellophane).

“Biomass” in Fig. 2A refers to a fresh (wet) weight. Considering the reviewers' comments, we conducted the additional measurements for both proteins and dry weight (Supplementary Fig. 5 in the revised manuscript). The results consistently show the

statistical significance on day 4, supporting our conclusion regarding Fig. 2A. The additional biomass measurement was also recommended by the reviewer #3. Although the statistically significant difference in biomass might appear minimal in Fig. 2A, this would be the matter of scale and not necessarily mean that it is not physiologically relevant. Throughout the manuscript, we have discussed the biological significance of *S. davawensis* CIS in relation to the colony morphology and developmental phenotype, and the effect on biomass accumulation is only one aspect of this complex phenotype. We do not consider that CISs directly regulate the central energy metabolism or essential synthetic machineries of *S. davawensis*. Rather, the biomass accumulation could be interpreted as a sign of the CIS effect on the colony morphology at the specific growth stage which we refer to as phase 1.

To Reviewer #2:

We greatly appreciate the time and the insightful comments on our manuscript. We are particularly grateful to the reviewer for the recognition of the manuscript's clarity and the novelty of our findings, and kindly mentioning “*The clarity of the manuscript is commendable, offering valuable insights into the regulatory role of CIS systems in Streptomyces development. Notably, it unveils, for the first time in these organisms, an effector associated with the CIS system*”. We have carefully considered all the comments and have thoroughly addressed all the feedback provided in the following section. Along with these, we have also made specific revisions to the manuscript and figures by refining the text or adding data to ensure clarity and readers' comprehensiveness. The changes on the manuscript are highlighted in blue.

Furthermore, we have expanded specific discussions to Supplementary Notes to provide deeper context and interpretation of the findings without exceeding the content of the main text. Point-by-point responses to the comments are as follows.

1) p.5 lines 6-17: The author conducted a phylogenetic analysis employing the CIS tube protein to assert the uniqueness of the *S. davawensis* CIS group. However, it's worth noting that CIS gene clusters typically encompass at least two tube proteins rather than a singular one. Is there experimental evidence validating the significance of this specific tube protein in the formation of CIS particles? In contrast, the manuscript and other articles robustly demonstrate the significance of the sheath protein. Utilizing the sheath protein for phylogenetic analysis would likely offer a more compelling argument.

We conducted a phylogenetic analysis for the CIS tube proteins because the amino acid sequences of tube proteins are relatively highly conserved compared with other structural proteins and their structural and evolutionary significance have been demonstrated in other CISs (Shikuma *et al. Science* 2014; Jiang *et al. Cell* 2019). As the reviewer pointed out, the numbers of CIS tube protein homologs can vary depending on the CIS gene clusters and the *S. davawensis* CIS gene cluster and its closer relatives (shown in Supplementary Fig. 1) would have a singular one. In our preliminary analysis, we have found that the *S. davawensis* CIS tube protein and its relatives consistently show the highest similarity with the tube proteins (often referred to as

Afp1/PVC1 homologs) encoded upstream of another one in other CIS lineages, suggesting that the *S. davawensis* tube protein shares an ancestor with the former tube proteins rather than the latter. Based on this, we chose the tube proteins with the highest similarities and constructed the phylogenetic tree.

According to the reviewers' suggestion, we constructed a tree for the CIS sheath proteins (Supplementary Fig. 1). As the numbers of the sheath proteins also vary depending on the CIS lineages (one to three per gene cluster), we collected the amino acid sequences with the highest similarities as described above. Overall, this tree is consistent with the tree for the tube proteins and would support our claim on the evolutionary uniqueness of the *S. davawensis* CIS.

2) p.5, line 16: The authors illustrate the uniqueness of *S. davawensis* CIS. It would be informative to elucidate the structural and elemental similarities or differences between *S. davawensis* CIS and well-known counterparts like *S. coelicolor* or *S. venezuela*. A schematic illustration of *S. davawensis* CIS could prove beneficial in aiding readers' comprehension.

Thank you for the suggestion. We added a comparison of the previously investigated *Streptomyces* CISs and *S. davawensis* CIS with respect to their structural proteins and putative effector proteins (Supplementary Fig. 19 and Supplementary Notes). These additional descriptions would emphasise the uniqueness of *S. davawensis* CIS within the genus. Furthermore, we also added a schematic illustration of *S. davawensis* CIS for better readers' comprehension (Supplementary Fig. 20).

3) The authors generated a mutant by introducing a stop codon at Gln78 of BN159_7569 (CisS), effectively preventing the translation of the sheath protein in CisS. However, in line 6 on page 6, it is stated that the granule structure consists of CISs composed of baseplate protein. How do the authors ensure that the translation of other genes downstream of CisS are not prematurely terminated? Have the author conducted analyses to detect the expression of other genes within this gene cluster?

Thank you for the comment. As Tme downstream of CisS was detected in the $\Delta cisS$ mutant by western blotting using anti-Tme-C antiserum, we consider that the Gln78*

mutation of *cisS* do not terminate the translation of the downstream CIS proteins (the result was attached below). Considering the reviewer's comment, however, we removed the words “mainly consisting of baseplate proteins” from the sentence to avoid specifying the composition of the granules and misleading readers (page 6, line 7 of the revised manuscript).

4) While I observe consistency between the trends asserted by the authors, it's noteworthy that the *P*-values in Figure 2 (G-I) do not consistently reach a "significant" threshold, thereby casting doubt on the validity of certain conclusions drawn. A similar observation can be made for Fig. 4K ($P=0.068$).

Given the concern which the reviewer has raised regarding Fig. 4K, we repeated the biomass measurement with increased sample numbers (5 per each strain) to evaluate a statistical significance more sensitively. As a result, we observed a distinguishable difference between the strains on day 4 with the *P* values = 0.000490.

As the main conclusion for Fig. 2G-I is that eDNA concentrations on day 2 reached $P = 0.05$, a widely accepted significance threshold, indicating the significant difference in ECM compositions between the wildtype and the $\Delta cisS$ mutant colonies at the time point, we believe that the other values with *P* values above the threshold would not affect the conclusion of the current study.

5) Have the authors investigated Programmed Cell Death (PCD) in connection to their CIS system, such as through the application of live/dead staining, a method previously employed in the study of other CIS systems in *Streptomyces*?

Before the initial submission, we have conducted a Live/Dead experiment employing Syto9/PI dye system to investigate whether PCD is implicated in the observed biological phenotypes or not. Although a recent study on *Streptomyces coelicolor*

claimed that a significant cell death associated with the CIS production was detected by the Syto9/PI system (Vladimirov *et al. Nature Communications* 2023), we did not observe any signs of the CIS-associated cell death in *S. davawensis*. In addition, the release of proteins into extracellular milieu, which must be facilitated by PCD and the subsequent cell lysis, was comparable between the wildtype strain and the CIS-deficient mutant (Fig. 2). Considering these results and the absence of a potential lytic effector(s) in the gene cluster, we concluded that the CIS-mediated PCD is unlikely at least in *S. davawensis*. Regarding the potential, detrimental consequence of the Tme action in *S. davawensis*, please see also the additional discussion in the section *1. Tme as an effector protein non-lethal in S. davawensis* in Supplementary Notes.

6) In lines 17-19 on page 6, the authors show that the mutant exhibits impaired multicellular development, specifically in aerial hyphae and spore formation. Is there a potential correlation between these morphological changes and the production of secondary metabolites? Has the author conducted a comparison of roseoflavin production between the wild type (WT) and the mutant to explore this aspect further?

We did not conduct a precise measurement for roseoflavin production. However, we could observe elevated secretion of a red pigment, which would be roseoflavin, in $\Delta cisS$ and *tme* Gln2006* mutants. We anticipate that the secondary metabolite production may have been altered in response to cell envelope stresses and nutrient depletion imposed by the abnormal aggregation of mycelia within the biofilm-like structures (Fig. 2). Although we did not investigate the phenomenon further and this would be out of scope of the current study, the CIS-associated change of antibiotics production may be interesting aspect of *Streptomyces* being worth future investigation.

7) Protein interactions play a significant role in supporting several conclusions made by the authors, as exemplified in lines 28-36 on page 7. The author utilized a pull-down assay to identify partners of the CIS tip protein. Were alternative methods employed to directly validate the interaction between BN159_7576 and the tip protein? Furthermore, in lines 6-8 on page 10, the authors employed AlphaFold to predict protein interactions between BN159_7588 and BN159_7587. However, given the artificial nature of AlphaFold predictions, additional evidence, such as bacterial 2-hybrid assays, should be provided by the authors to strengthen their claims.

Thank you for the suggestion. Before the initial submission, we have tried to directly detect the interaction between the spike complex and Tme. However, all attempts were hampered by highly aggregative nature of BN159_7578 and BN159_7580. Although we tested both *in vivo* and *in vitro* expression of these proteins, we could not obtain the proteins enough to detect the protein-protein interaction. We assume that an unannotated protein(s) encoded within the *S. davawensis* CIS gene cluster might assist the proper assembly of the spike complex, which would then interact with Tme. Furthermore, it was impossible to isolate the predicted disordered region of Tme presumably important for the spike-Tme interaction, due to its highly unstable nature. Tapemeasure protein of the *Staphylococcus aureus* phage 80 α , that Tme has a remote homology, seems to interact with the Dit-Tal complex at its disordered region (PDB: 6V8I) corresponding to the boundary region between the disordered region and predicted hydrophobic helices in the Tme polypeptide (Fig. 4a). Therefore, successful isolation of this segment would be needed to construct a direct detection system for the Tme-spike interaction. Due to these technical limitations, we combined the pulldown of CIS spike complex and quantitative proteomic analysis to detect the effector-spike interaction in an indirect manner. As we have shown in a previous study (Nagakubo *et al. mSphere* 2023) and the current manuscript, this approach would be effective to identify CIS-associated proteins especially when the interaction partners are structurally unstable, and it can be applied to future studies focusing on CIS effectors which direct interaction assays are technically inapplicable. Elucidation of a detailed interaction mechanism of Tme and other CIS structural proteins will be our next challenge.

We provide the additional data indicating the interaction between BN159_7587 (putative fiber) and BN159_7588 (peptidoglycan binding protein) in the revised manuscript (Supplementary Fig. 16c) and attached it below. We prepared the lysates of *E. coli* cells expressing 3 \times FLAG-BN159_7587 and then performed the pull-down assay with BN159_7588-His₆ that has already been shown to be purified by Ni²⁺-affinity chromatography in the initially submitted manuscript. Coelution of 3 \times FLAG-BN159_7587 with BN159_7588-His₆ from a Ni²⁺-affinity chromatography column was confirmed by western blotting using anti-FLAG antibody. This result would support the AlphaFold2-predicted interaction between these proteins.

8) The assertion in line 19 on page 10, suggesting that CIS particles are unaffected, lacks convincing support. Figure 2H is inadequately detailed, making it challenging to draw a meaningful comparison between the CIS particles of the *tme* Q2006 mutant and the wild type for a conclusive determination.

Considering the reviewer's comment, we modified the related sentence to provide more precise interpretation of the result (page 10, lines 35-36). Detailed structural analysis employing cryo-EM would be necessary for a strong assertion. In addition, we replaced the image with clearer one.

9) In lines 10-12 on page 10, the pivotal conclusion regarding the ability of Tme to degrade genomic DNA requires additional substantiation. The author introduced the Tme-C H23A point mutation, but the evaluation was limited to the nucleolytic activity on the pUC19 plasmid. Why did the author exclusively examine this plasmid and not include genomic DNA from *S. davawensis*? Is there a possibility that this point mutation may not effectively impede the nucleolytic activity of Tme-C on *S. davawensis* genomic DNA? Furthermore, has the author explored whether Tme-C can degrade RNA from *S. davawensis*? Additional experimentation is needed to address these aspects comprehensively.

According to the suggestion, we added the results of the incubation of Tme-C (H23A) and genomic DNA. Consistent with the result on pUC19 and the bioinformatic prediction, the mutation on the catalytic histidine abolished the nucleolytic activity toward genomic DNA as well as the plasmid. Furthermore, we also added the experimental result showing that Tme-C can degrade RNA as well as DNA (Supplementary Fig. 12e). This result is consistent with the remote homology of Tme with *S. pneumoniae* EndA that belongs to the DNA/RNA non-specific endonuclease family. Related descriptions were added to the revised manuscript (page 9, lines 16 and 21).

10) Does the expression pattern of Tme align with that of the CIS system during the development of *S. davawensis*? If not, what mechanisms does *S. davawensis* employ to safeguard itself from genome degradation by Tme?

We did not find the differential expression of Tme and the CIS system in *S. davawensis*. We assume that *S. davawensis* is able to circumvent the potential, detrimental consequence of the Tme action because of (i) low nucleolytic activity of the Tme C-terminal domain (Tme-C), (ii) spatial sequestration of Tme inside a CIS particle until ejection, and (iii) the multinuclearity of *Streptomyces* mycelia and the hydrophobic nature of Tme. Detailed explanations are in the section *1. Tme as an effector protein non-lethal in S. davawensis* in Supplementary Notes.

To note, despite our effort before the initial submission, we could not find out an immunity protein which antagonises Tme. We have constructed co-expression systems in *E. coli* for Tme and each of unannotated proteins encoded in the CIS gene cluster, but none of these proteins alleviate the cytotoxicity of the overexpressed Tme to *E. coli*.

To Reviewer #3:

We greatly appreciate the time and the insightful comments on our manuscript. We are particularly grateful to the reviewer for the recognition of the novelty of our findings. We have carefully considered all the comments and have thoroughly addressed all the feedback provided in the following section. Along with these, we have also made specific revisions to the manuscript and figures by refining the text or adding data to ensure clarity and readers' comprehensiveness. During these additional works, we have also developed new rabbit antibody to provide a supplementary western blotting result. Furthermore, we have expanded specific discussions to Supplementary Notes to provide deeper context and interpretation of the findings without exceeding the content of the main text. The changes on the manuscript are highlighted in blue. Point-by-point responses to the comments are as follows.

Why do the authors exclude the possibility that Tme-dependent cell lysis (since no immunity is there?) could result in DNA release in the extracellular environment, instead of resulting from a pore-forming and DNA cleavage activity in targeted cells? Again, I may have missed a point here. Also, why don't the authors discuss the presence or absence of an immunity protein? It may well not be necessary in this particular system, but it would be nice to explain why.

Thank you for the suggestion. We assume that *S. davawensis* is able to circumvent the potential, detrimental consequence of the Tme action because of (i) low nucleolytic activity of the Tme C-terminal domain (Tme-C), (ii) spatial sequestration of Tme inside a CIS particle until ejection, and (iii) the multinuclearity of *Streptomyces* mycelia and the hydrophobic nature of Tme. Detailed explanations are in the section *1. Tme as an effector protein non-lethal in S. davawensis* in Supplementary Notes. We also added the related description to the Discussion (page 13, lines 9-11).

To note, despite our effort before the initial submission, we could not find out an immunity protein which antagonises Tme. We have constructed co-expression systems in *E. coli* for Tme and each of unannotated proteins encoded in the CIS gene cluster, but none of these proteins alleviate the cytotoxicity of the overexpressed Tme to *E. coli*. We

thus consider that an immunity protein would not be encoded at least in the gene cluster. However, we did not describe this point in the manuscript as it would require an extensive genome-wide experiment and structural analysis, which will be worth another full paper, for a conclusive statement.

Section starting P6 L22:

How was biomass measured? I am not convinced by the measurements showing a decreased growth in phase 1, as other differences mentioned could possibly affect these measurements. In addition, although it is statistically significant, it appears minimal and only shown in one measure. Such claim should be backed up with more points. Anyway, is it relevant at all?

Biomass was measured by wet weight of the colonies grown on a cellophane membrane. For more clarity, we modified the related sentence in the Methods section (page 15, lines 10-13). Considering the reviewers' comment, in addition, we conducted the additional measurements for both proteins and dry weight (Supplementary Fig. 5 in the revised manuscript). The results consistently show the statistical significance on day 4, supporting our conclusion regarding Fig. 2a.

Although the statistically significant difference of biomass might appear minimal in Fig. 2a, this would be the matter of scale and not necessarily mean that it is not physiologically relevant. Throughout the manuscript, we have discussed the biological significance of *S. davawensis* CIS in relation to the colony morphology and developmental phenotype, and the effect on biomass accumulation is only one aspect of this complex phenotype. We do not consider that CISs directly regulate the central energy metabolism or essential synthetic machineries of *S. davawensis*. Rather, the biomass accumulation could be interpreted as a sign of the CIS effect on the colony morphology at the early growth stage which we refer to as phase 1.

Fig. 4G: the results do not clearly show if the Tme Q2006* protein is produced at all. Hence, the authors cannot specifically conclude that Tme nuclease activity is responsible for the phenotypes observed (even though I am totally convinced Tme is required).

We appreciate the comment. Since receiving the review report, we have developed rabbit antiserum using a peptide (RRRKRKERAALKSRTPEPKN) corresponding to N-terminal region of Tme (18-36 aa) and conducted western blotting. The signals of the native Tme and slightly smaller Tme Gln2006* were detected in the mycelial lysates of the wildtype strain and the *tme* Gln2006 mutant, respectively, which are the same lysates as used in Fig. 4g (Supplementary Fig. 17c). The additional data would complement the results and discussions regarding the *tme* Gln2006* mutant.

Sup Fig. 5; I am having a hard time believing how the authors interpret the results or the result itself. First, the picture may have been contrasted excessively, therefore hiding the smear gDNA should show. Yet, wouldn't we expect Tme activity to results is a more pronounced DNA smear. Here it would be as if gDNA is cut at specific sites.

We consider that the degradation of genomic DNA by Tme-C would not be sequence-specific as we could not find any specific cut sites in a NGS analysis of the Tme-C-degraded genomic DNA conducted in prior to the initial submission. Also, the non-specific nucleolytic activity is a common feature of EndA-related endonucleases with which Tme-C has the remote homology. We have already shown the Tme-C-degraded genomic DNA in different pixel intensity settings in Supplementary Fig. 7, and the smear DNA was not observed in both images. This result would indicate that the random degradation of genomic DNA by Tme-C is limited to a partial level due to the significantly low hydrolytic activity of the enzymatic domain, generating a similar size of DNA fragments. We have added the related descriptions regarding the low Tme-C activity in the section 4. *Unique enzymatic characteristics of Tme-C* in Supplementary Notes. This additional section is also related to the response to the reviewer's comment regarding Fig. 4F below.

Other specific comments:

L10: wording is ambiguous. Are all three processes decreased or only DNA release, while biomass accumulation and spore formation are promoted.

We appreciate the comment. We changed “decreased” to “decreases in” for clarity (page 2, line 10).

L13: it would be more interesting to discuss the diversity of molecular activities associated with CISs so far rather than just vaguely mentioning “few bacterial species”

We appreciate the suggestion. We added the related sentences to the end of the paragraph (page 3, lines 17-24). This revision would make the introduction more informative.

P3, paragraph starting L28, more specifically P4 Lines 4: The way “host” is used is kind of confusing.

We appreciate the comment. To avoid confusion, we removed the word “host” or replace it with “producer bacteria” throughout the paragraph.

P6 L1: in which conditions were the lysates prepared?

The lysates were prepared as described in the section *Extraction of CIS particles* in the Methods.

P6 L15: Fig 1E-F and not 1D-F?

We apologise for the confusion. We corrected “Fig. 1D-F” to “Fig. 1E and F”.

P6, Paragraph L10: it is not clear to me why grow rate was not compared first, and only mentioned in the next paragraph.

The structure of the paragraphs reflects experimental procedures employing *S. davawensis*. Typically, our biological experiments employing *Streptomyces* species start with the isolation of spores that are then inoculated into media or directly subjected to various analyses. Due to the aggregative and multicellular morphology of *Streptomyces*, in our case precise and reproducible results were achieved by inoculating unicellular

spores rather than multicellular mycelia. Therefore, in the current study, we first observed and examined spore formation before growth measurement when investigating the *Streptomyces davawensis* phenotypes. This is the reason why the growth measurement of *S. davawensis* follows the observation of spore formation in our manuscript. To clarify that we first observed spore formation and these spores were used for further analyses, we modified the related sentences (page 6, lines 11-13 and 27).

Figure 4F: can the authors better explain why there is a light upper band when pUC19 was incubated with Tme-C please? Do they mean that the His23Ala mutant cannot bind DNA? Is it known if this residue is required for DNA binding? Also, the Materials and Methods section does not give precision, but I am amazed that Tme was so easily produced and purified. Yet, Fig 4C clearly shows cytotoxicity of Tme produced in BL21 cells. Maybe I missed a particular detail.

We believe that the nucleolytic activity and cytotoxicity of Tme-C is very weak compared with other EndA-related endonucleases and therefore we could have isolated this enzymatic domain from the *E. coli* cells. Indeed, the band of 2.5 ng/μL of pUC19 disappeared after incubation with 50 ng/μL of Tme-C at 30°C for 180 min, suggesting that the Tme-C-catalysed hydrolysis is very slow compared with EndA (please see also the section 4. *Unique enzymatic characteristics of Tme-C* Supplementary Notes). Please note that the cytotoxicity caused by Tme/Tme-C was observed when these proteins are overexpressed with the extremely strong T7 RNA polymerase/T7 promoter system in *E. coli*, and some cells can grow slowly even after the induction due to the low activity of Tme-C.

We have several data implying that the Tme-C activity is limited to a low level. Details are described in the section 4. *Unique enzymatic characteristics of Tme-C* in Supplementary Notes. In brief, Tme-C appears to lack several amino acid residues that facilitate hydrolysis of the DNA phosphodiester in *S. pneumoniae* EndA. In addition, Tme-C can form a stable, probable covalent complex with dsDNA substrate, which could be a rate-limiting step in the Tme-C-catalysed nucleolytic reaction and lead to the upward shift of the DNA band in agarose gel. Regarding this, we provide additional experimental results in Supplementary Fig. 13. According to the addition of these data,

we modified the related sentences and added new descriptions to the Results section (page 9, lines 6-36; page 10, line 1).

P12 L1: Can the authors provide any data to support CPP-like properties? Did they use CPP detection tools available? The whole Tme-pore hypothesis seems a bit bold. The pore may be provided by a receptor protein interacting with Tme.

Thank you for the comment. We are currently unable to directly demonstrate CPP-like properties of the arginine-rich segment of Tme due to difficulties in synthesis, and therefore this is hypothetical. We used a web-available tool C2pred to predict the potential CPP-like properties of the segment.

According to the reviewer's comment, we added the description regarding the possible involvement of a receptor protein to provide another hypothesis for the Tme function (page 12, line 34).

Reviewer #1 (Remarks to the Author):

The new version of the manuscript addressed all my doubts and concerns. I recommend the publication of the manuscript in Nature Communications. Congratulations to the authors for the nice manuscript.

Reviewer #2 (Remarks to the Author):

We appreciate the effort that the authors made in improving the manuscript. The authors now added additional data and included a proposed model for functioning of the CIS. I find two things related to the new data difficult to interpret. The authors now show in Supplementary Figure 12e robust activity of Tme-C in degrading RNA, suggesting that RNA behaves more like a natural substrate for Tme-C compared to gDNA. This finding also appears to contradict the author's statement regarding Tme-C's low nuclease activity. The author illustrated through Western blot analysis (rebuttal for comment 3) that deletion of *cisS* did not terminate the translation of Tme. Considering its robust RNA-degrading activity and low gDNA-degrading activity, it's difficult to understand how the $\Delta cisS$ mutant survives from RNA elimination by Tme. Furthermore, how does this nuclease function come back in the model (Fig. S20)? I appreciate having a proposed model, but find it difficult to interpret it based on the data shown in the paper. How is the CIS system ending up on the outside? Is this due to lysis? And how does it then bind to the cell wall (no direct evidence is provided in the paper). And is the effector injected back into the cells? For what reason? One small other comment is to replace Fig. 1B with Fig. S1. The majority of the paper deals with the sheath protein, and it is quite strange to then show the phylogenetic tree based on the tube protein (also because the major conclusions are the same).

Reviewer #3 (Remarks to the Author):

The authors do address all of my concerns but one (see below). This does not change the conclusions drawn by the authors but I would not use such data myself in any paper. Otherwise there is no doubt this new version of the manuscript is much improved.

My initial comment was:

Fig. 4G: the results do not clearly show if the Tme Q2006* protein is produced at all. Hence, the authors cannot specifically conclude that Tme nuclease activity is responsible for the phenotypes observed (even though I am totally convinced Tme is required).

The authors trying to address it:

We appreciate the comment. Since receiving the review report, we have developed rabbit antiserum using a peptide (RRRKRKERAASRTPEPKN) corresponding to N-terminal region of Tme (18-36 aa) and conducted western blotting. The signals of the native Tme and slightly smaller Tme Gln2006* were detected in the mycelial lysates of the wildtype strain and the *tme Gln2006* mutant, respectively, which are the same lysates as used in Fig. 4g (Supplementary Fig. 17c). The additional data would complement the results and discussions regarding the *tme Gln2006** mutant.

My new comment:

The Western blot shown in Supplementary Fig. 17c (labelled 17b in the text) does not bring any improvement and does not do justice to the quality of the whole study. I would not be able to draw any conclusion from it if I did not know the result in advance. At least, the authors need to provide a negative/specificity control if they want to use this data. Also, Fig. 4g is not used anymore in the new version of the text.

To Reviewer #2:

Thank you very much for reviewing the revised manuscript and providing us with the insights to further improve it. We have carefully considered the comments and addressed them as follows.

[Reviewer's comment]

We appreciate the effort that the authors made in improving the manuscript. The authors now added additional data and included a proposed model for functioning of the CIS. I find two things related to the new data difficult to interpret. The authors now show in Supplementary Figure 12e robust activity of Tme-C in degrading RNA, suggesting that RNA behaves more like a natural substrate for Tme-C compared to gDNA. This finding also appears to contradict the author's statement regarding Tme-C's low nuclease activity. The author illustrated through Western blot analysis (rebuttal for comment 3) that deletion of *cisS* did not terminate the translation of Tme. Considering its robust RNA-degrading activity and low gDNA-degrading activity, it's difficult to understand how the Δ *cisS* mutant survives from RNA elimination by Tme.

[Response to the comment]

We appreciate the insightful comment. We consider that the ability of Tme-C to degrade RNA would not necessarily mean that it is a “natural” substrate for Tme-C as in biochemical enzymology *in vitro* activity of an enzyme is not often sufficient to determine its natural substrate. For determining natural substrates for enzymes, it would be necessary to at least identify a significant biological phenotype associated with the “natural” enzyme-substrate pair. Regarding this, we have found the significant, Tme-associated difference in the release of gDNA in *S. davawensis*, leading to the hypothesis that CIS and its cognate effector Tme affect eDNA release. As this is the reason why we have focused on the Tme activity toward DNA rather than RNA, the RNA-degrading activity of Tme-C would not be the main point of the current study. In addition, we believe that the RNA-degrading activity of Tme-C is also low enough to circumvent significant RNA elimination in *S. davawensis* which can potentially lead to cell death. If the RNA-degrading activity of water-soluble Tme-C was as high as those of typical RNase effectors, overexpression of Tme-C in *E. coli* should markedly decrease the cell viability by eliminating RNA vulnerable to the nucleolytic activity. However, the toxic effect of the overexpressed Tme-C on the *E. coli* cells was rather moderate, and the native Tme-C can be isolated from the Tme-C expressing cells, suggesting that the *in vivo* nucleolytic activity of Tme-C toward both DNA and RNA is not highly detrimental to bacterial cells (Fig. 4d). Furthermore, it is notable that Tme is not toxic to *E. coli* at a low expression level. In Fig. 4c, the *E. coli* strain harbouring pET26b::*tme* showed similar growth with the strain harbouring the non-toxic mutant Tme in the absence of an inducer IPTG,

indicating that Tme at a basal expression level do not cause significant toxicity to the *E. coli* cells. Please note that any systems such as T7 lysozyme suppressing the leaky expression were not employed in this study. Given that the leaky expression of bactericidal proteins cloned into pET vectors with the extremely strong T7 promoter/T7 RNA polymerase system often results in significant death of the host cells, Tme seems unlikely to act as a “toxin” at a basal expression in the pET system or possibly endogenous expression level within *S. davawensis* mycelia. Additionally, since the Tme polypeptide containing probable disordered regions is predicted to be unstable in solution (instability index (II) is 46.01, above instability threshold 40), it can be speculated that, in the absence of a CIS particle that can load Tme inside its lumen, Tme might be folded improperly in aqueous cytosol or potentially form an inactive complex with itself and/or other proteins (van der Lee *et al. Chemical Reviews* 2014 <https://doi.org/10.1021/cr400525m>). The predicted structural instability of Tme in aqueous cytosol crowded with macromolecules may prevent delocalised Tme from fully exhibiting its activity, further ensuring that the producer bacterium *S. davawensis* circumvents the potential, detrimental consequence of the nucleolytic activity of Tme. Again, as we responded previously, we could not detect any signs of CIS/Tme-associated cell death in this bacterium and could not find out an immunity protein that antagonises the Tme activity and is encoded in the CIS gene cluster. Our results would thus suggest that there may be some exceptions to the previously proposed model of DUF4157 domain-containing CIS effectors as cell-killing toxins (Geller *et al. Nature Communications* 2021) and would expand the concept of the functional diversity of the intriguing class of CIS effectors. Considering the reviewer's comment, we added the related descriptions about to the section *Tme as an effector protein non-lethal in S. davawensis* in Supplementary Notes (highlighted in blue).

[Reviewer's comment]

Furthermore, how does this nuclease function come back in the model (Fig. S20)? I appreciate having a proposed model, but find it difficult to interpret it based on the data shown in the paper. How is the CIS system ending up on the outside? Is this due to lysis? And how does it then bind to the cell wall (no direct evidence is provided in the paper). And is the effector injected back into the cells? For what reason?

[Response to the comment]

Thank you for the comment. Although the schematic model of the *S. davawensis* CIS (Supplementary Fig. 20 in the previous version) was relying on the previously proposed models of extracellular CISs (e.g. Jiang *et al. Cell* 2019) and the data presented in our manuscript (Supplementary Fig. 15 and 16), we agree that some details of this model remain speculative. Considering the reviewer's comment, we decided to remove the schematic illustration from the manuscript to avoid confusion. We will elucidate

more detailed action mechanism of the *S. davawensis* CIS in future papers.

[Reviewer's comment]

One small other comment is to replace Fig. 1B with Fig. S1. The majority of the paper deals with the sheath protein, and it is quite strange to then show the phylogenetic tree based on the tube protein (also because the major conclusions are the same).

[Response to the comment]

Thank you for the suggestion. We replaced Fig. 1b with Supplementary Fig. 1.

To Reviewer #3:

Thank you very much for reviewing the revised manuscript and providing us the suggestion to further improve it. We also apologise for the confusing typographical error in labelling the panel. For the additional revision, we replaced the western blot image with a clearer one including a negative control and added new data showing the reactivity and specificity of the newly developed antiserum. Details are described below.

[Reviewer's comment]

The Western blot shown in Supplementary Fig. 17c (labelled 17b in the text) does not bring any improvement and does not do justice to the quality of the whole study. I would not be able to draw any conclusion from it if I did not know the result in advance. At least, the authors need to provide a negative/specificity control if they want to use this data. Also, Fig. 4g is not used anymore in the new version of the text.

[Response to the comment]

We appreciate the suggestion. For the additional revision, we replaced the western blot image with a clearer one including control samples and attached them below for reference (Supplementary Fig. 17c and d in the revised manuscript). Although the detection of Tme remains challenging presumably due to the large size, the presence of the highly hydrophobic or positively charged regions, and low expression level, we did our best to optimise the protocol for the newly developed antiserum against the N-terminal region of Tme (Tme-N). Specifically, we have examined antisera collected from a rabbit at different time points (7 or 14 days post the 4th administration of the immunogen peptide), separation gels (isocratic gels with different acrylamide concentrations and a gradient gel), membranes (PVDF membranes with different hydrophobicity), and various conditions for transblotting (Tris-glycine systems with different compositions and Trans-Blot Turbo system purchased from Bio-Rad), blocking/antibody binding (skim milk and several commercially available blocking/binding reagents), and chemiluminescence reaction as possible as we can. We modified the Methods section to clarify the methodological differences between the experiments (highlighted in blue; lines 1-18, page 20). In addition, we added the data of an *E. coli* strain expressing recombinant Tme-N showing the reactivity of anti-Tme-N serum toward Tme (Supplementary Fig. 17c in the revised manuscript, red asterisks). In the new blot image of the *S. davawensis* strains, the bands above 200 kDa were consistently observed in the lysates of both the wildtype strain and the *tme* Gln2006* mutant (Supplementary Fig. 17d in the revised manuscript). Although we have failed to construct a mutant lacking the N-terminal

region of Tme despite considerable effort, we interpret these bands as native Tme and Tme Gln2006* for the following reasons: (i) the corresponding band was not detected in the lysate of *Streptomyces lividans* TK23, a close relative of *S. davawensis* lacking a Tme homolog, (ii) the difference in the apparent sizes of the bands would be consistent with the difference in their theoretical sizes (native Tme, 242.8 kDa; Tme Gln2006*, 216.2 kDa) while the apparent sizes of the other (non-specific) bands were the same between the strains, and (iii) the recombinant Tme-N, containing the target epitope present in both native Tme and Tme Gln2006*, was detected by anti-Tme-N serum (Supplementary Fig. 17c in the revised manuscript). The seemingly, slightly larger sizes of Tme and Tme Gln2006* than their theoretical sizes could be explained by the presence of the highly positively charged CPP-like region within the polypeptides, which potentially delay the migration of the polypeptides toward the anode during SDS-PAGE. Although signal intensity of the bands of Tme in Supplementary Fig. 17 seems to be not as strong as those detected by anti-Tme-C serum, we assume that this is due to the differential reactivity of the antibodies. Particularly, anti-Tme-C serum was collected from a rabbit which has died after the 4th administration of the immunogen peptide and the antiserum showed markedly high reactivity to the immunogen peptide in a preliminary ELISA assay. As this would be a rare case, it is currently impossible to obtain anti-Tme-N antibody with the comparable reactivity toward Tme.

We believe that these data would indicate the expression of the native Tme and its Gln2006* mutant in the *S. davawensis* strains used in the current study. Given the lack of a mutant with deletion of the N-terminal region of Tme due to a technical limitation, however, we modified the related descriptions in the manuscript to avoid a strong assertion (line 4, page 10; line 8, page 11).

Fig. 4g was referred to in line 35 on page 10 in the previous version of the main text.

Reviewer #2 (Remarks to the Author):

The new version of the manuscript addressed all my doubts and concerns. Congratulations to the authors for the nice manuscript.

Reviewer #3 (Remarks to the Author):

This new version addresses my previous concern. Therefore I can only acknowledge the beautiful work and recommend publication.